# Analysis of Cyber Security Attacks and Its Solutions for the Smart grid Using Machine Learning and Blockchain Methods

Tehseen Mazhar [1,*], Hafiz Muhammad Irfan [2], Sunawar Khan [2], Inayatul Haq [3], Inam Ullah [4], Muhammad Iqbal [5] and Habib Hamam [6,7,8,9,*]

1   Department of Computer Science, Virtual University of Pakistan, Lahore 51000, Pakistan
2   Department of Computer Science, Islamia University Bahawalpur, Bahawalnagar 62300, Pakistan
3   School of Information Engineering, Zhengzhou University, Zhengzhou 450001, China
4   BK21 Chungbuk Information Technology Education and Research Center, Chungbuk National University, Cheongju 28644, Republic of Korea
5   Institute of Computing and Information Technology, Gomal University, Dera Ismail Khan 29220, Pakistan
6   Faculty of Engineering, Université de Moncton, Moncton, NB E1A3E9, Canada
7   Spectrum of Knowledge Production & Skills Development, Sfax 3027, Tunisia
8   International Institute of Technology and Management, Commune d'Akanda, Libreville 1989, Gabon
9   Department of Electrical and Electronic Engineering Science, School of Electrical Engineering, University of Johannesburg, Johannesburg 2006, South Africa
*   Correspondence: tehseenmazhar719@gmail.com (T.M.); habib.hamam@umoncton.ca (H.H.)

**Abstract:** Smart grids are rapidly replacing conventional networks on a worldwide scale. A smart grid has drawbacks, just like any other novel technology. A smart grid cyberattack is one of the most challenging things to stop. The biggest problem is caused by millions of sensors constantly sending and receiving data packets over the network. Cyberattacks can compromise the smart grid's dependability, availability, and privacy. Users, the communication network of smart devices and sensors, and network administrators are the three layers of an innovative grid network vulnerable to cyberattacks. In this study, we look at the many risks and flaws that can affect the safety of critical, innovative grid network components. Then, to protect against these dangers, we offer security solutions using different methods. We also provide recommendations for reducing the chance that these three categories of cyberattacks may occur.

**Keywords:** smart grid; cyber security; cyberattacks; machine learning; deep learning; data mining





## 1. Introduction

Modern technologies were integrated into the traditional electrical infrastructure to create a "smart grid". A smart grid has several ways to control operations and power. Examples of operational and energy measures include smart meters and appliances installed at the client's site, a production meter, renewable energy generators, smart inverters, and resources installed at the grid's location for energy efficiency [1]. Renewable energy generators can lower energy costs because it is free to produce energy from renewable sources, even though it is not always available and depends on variables like temperature, humidity, wind speed and direction, and location. Solar energy is influenced by the sun's brightness, cloud cover, and temperature [2]. The power that can be taken from the wind depends significantly on its direction and speed. Using renewable energy effectively and on time is possible because of the many technologies available for forecasting wind, solar, and battery state of charge. Sensors may communicate to and receive data from the smart grid because it has data transmission and reception capabilities [2]. These sensors provide data packets to the grid continuously. These data packets could include information on the production. Information on energy generation, use, voltage, and frequency may be found. The battery management system is vulnerable to hackers due to the communication channel used by existing battery-integrated grids to convey charge status. Batteries that are

overcharged or undercharged could become worthless as a result of cyber risks [3]. Figure 1 shows the components of a power grid that houses electrical support systems.

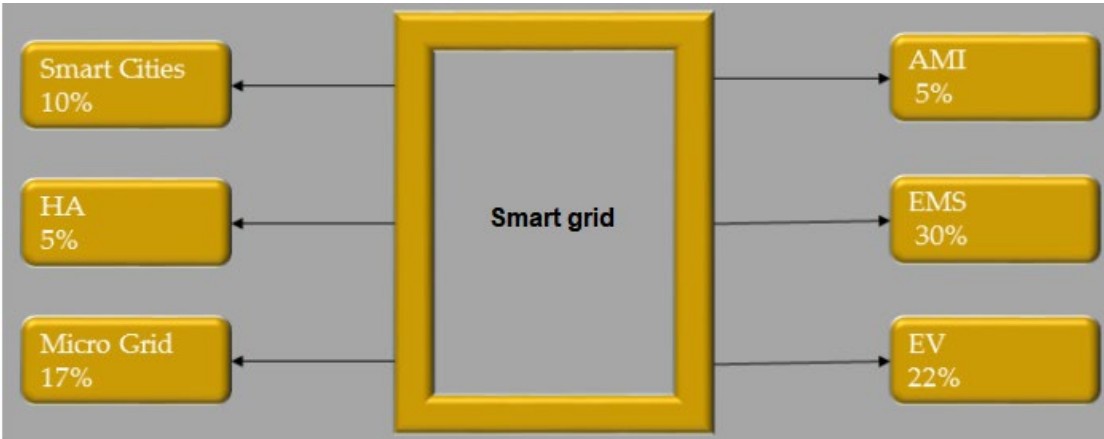

**Figure 1.** Components of a power grid that houses electrical support systems [2,3].

The smart grid has many benefits over traditional grids, such as better power quality, self-healing, cost-effectiveness with the integration of renewable energy, adaptive energy generation, more environmentally friendly operation, aggregation of distributed energy resources, real-time energy consumption monitoring at the customer end, integration of AI models to automate tasks, remote energy motoring, rapid response to faults, remote fault location. Smart grids are more attractive than conventional grids because of these benefits. The two most important problems are complexity and cybersecurity. It is more challenging to fix these vulnerabilities when smart grid data is stored on the cloud [4]. In addition to physical security, cybersecurity is a crucial element of the smart grid since it ensures its dependability and safety at all times. Not only are smart grids required to have cyber security, but [5] also shows that non-smart and older grids are susceptible to hackers. This study, shown in [5], shows how the power grid is affected when criminal software manages the whole power consumption of computers, including the CPU, GPU, hard drives, screen brightness, and laser printers. The study found that 2.5 to 9.8 million illnesses can potentially upset the system. Another study [6] found that when an attacker gains access to the IoT botnet for high-power smart appliances, it can lead to frequency instability, line failure, and increased operational costs. These kinds of attacks have the power to cause widespread shortages by manipulating energy consumption. As the grid's complexity rises, the likelihood of issues increases. Power networks, which are already noteworthy in and of themselves, are undergoing considerable changes due to the development of renewable energy sources, quick signal processors, and sophisticated sensors. These changes are severely disrupting the industry. These modifications have a considerable impact on the grid. Due to the existing situation, electricity producers and consumers must share information in both directions. A smart grid, which can dynamically monitor and regulate energy flow to deliver constant electricity for clients, is replacing the existing power infrastructure [7]. Data from research that have been published that deal with SG are shown in Figure 2.

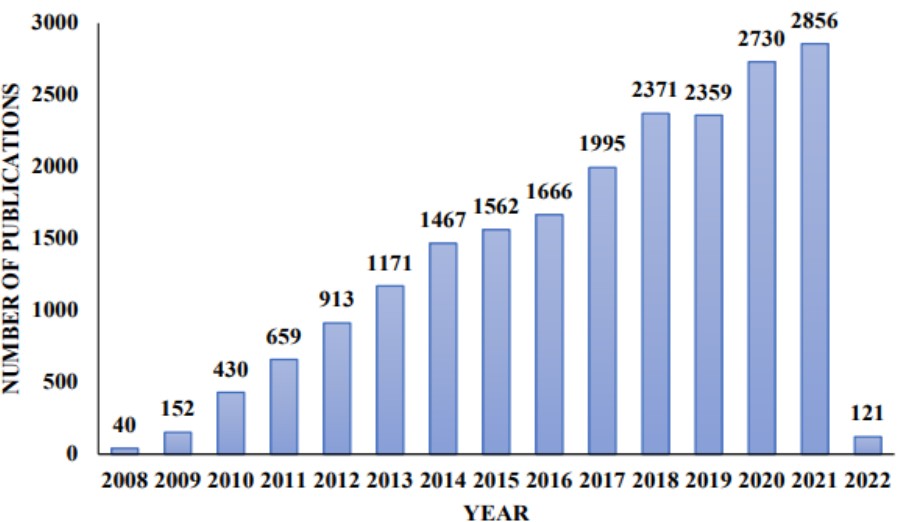

**Figure 2.** Publication statistics on SG [8].

**Table 1.** Existing Surveys Related to the Cyber-security of Smart grids.

| References | Cyberattacks | Objectives |
|---|---|---|
| [10] | Multiple cyberattacks were launched targeting the CIA computers and the five OSI communication layers. | The various forms of cyberattacks and the over-all necessity of taking prevention achievement. An analysis of multiple cyberattacks, including the requirements for their protection, as well as the directions for the future. |
| [11] | Analysis of traffic, social engineering, scanning an IP address, scanning a port, scanning a vulnerability, worms, denial of service attacks, forward data thefts, replays, violations of privacy, and DDoS. | Cyber-physical security of smart grids and potential attack scenarios based on information technology. Methods of prevention and detection, as well as the difficulties involved, concerning the threats posed by smart grids. |
| [12] | Attacks against the generation system, attacks against the transmission system, attacks against the distribution system and the client side, and attacks against the electrical market. | Critical cyber-physical attacks and the various ways to defend against them. Investigating the effects of combined cyber and physical attacks on smart grids. |
| [10] | DoS/DDoS attacks | The smart grid and all of its core elements. Methods now in use for various communication protocols and their underlying systems Attacks of the DoS and DDoS variety, and the effects they have on smart grids. |
| [13] | Some of the hacking techniques covered in this article are traffic analysis, social engineering, scanning IP addresses, monitoring ports, scanning vulnerabilities, worms, Trojan horses, DoS, FDI, replay, privacy violations, integrity violations, backdoors, MITM, jamming, popping the HMI, and masquerade. | Major cyberattacks against the smart grid and the effects have various security approaches to solve the cyber-security problem in smart grids. |
| [14] | Various forms of online attacks on confidentiality, integrity, availability, authorization, and authenticity. | The most commonly encountered challenges when dealing with smart homes and smart grids. A variety of cyberattack situations, each with its unique defensive measures. Strategies to protect against or avoid the occurrence of cyberattacks. |
| [1] | MITM, jamming, FDI, spoofing, DoS, malware, replay attacks. | Multiple cyberattacks have been directed at smart grids and the security systems used. |
| [15] | Attacks of various forms launched against energy corporations, renewable energy resources, and metering networks. | Vulnerabilities in the traditional electricity network that cyberattacks can target. In the case of smart grid metering networks, security, and privacy criteria must be addressed research in the future, including its trends and problems. |

Machine learning, deep learning, Data mining, evolutionary algorithms, fuzzy logic, and other similar techniques are all included in artificial intelligence. Machine learning is becoming increasingly important to researchers for danger detection. The authors of [16] used machine learning methods such as random forest, support vector machine, and neural networks to recognize jamming attacks. Their numerical tests show that the suggested random forest strategy works well. The authors employed machine learning techniques to identify social engineering attacks. The system uses unsupervised learning, so it doesn't need to be familiar with cyberattacks to recognize them. The authors examined different machine learning applications' accuracy, speed, and consistency. They discovered that support vector machines outperform competing strategies using computer simulations [17].

The authors of [18] used machine learning methods to protect against network-layer brute force attacks on the Secure Shell protocol. The authors developed scalable detection models with the help of classifiers like K-Nearest Neighbors decision trees and Naive Bayes that may be effective at making predictions. The author of [19] describes a different experiment that utilized machine learning. The idea of "first difference" from statistics and economics inspired the authors of this study to develop a classifier that can identify dangers to network time synchronization. They found that Artificial Neural Networks outperformed traditional techniques for detecting network security issues. An ANN model was used to identify MITM assaults, and the authors noted a high detection rate. The authors of [20] used machine learning techniques to identify and remove hackers from smart grids. The simulations conducted for this study showed that the suggested approach might have a high detection rate.

Deep learning has also been used to track cyberattacks on the smart grid. For example, the authors of [21] created a deep neural network and a deep learning ensemble technique based on decision trees. Ten-fold cross-validation was employed to assess the model. The evaluation results show that the suggested model beats the most effective methods currently available, such as random forest, Ada Boost, and DNN [22].

Cyberattacks on the smart grid can potentially be discovered through data mining, a type of AI. The authors of [20] discussed past research that used data mining techniques to spot fake data injection attacks in smart grids. These methods allow you to explore data patterns that you usually wouldn't be able to see and find ways in vast amounts of data. In [23], the authors used the data mining method known as Common Path Mining to find FDIA in their networks. To describe how the samples were arranged, they chose to use the idea of a "route." Every unique incident has a different course that has a wide range of flaws. A sequence is considered an attack if it fits within one of the paths. A Casual Event Graph can be used by the authors of [24] to identify FDIA in smart grids.

The training of historical datasets is the primary goal of the data mining techniques used hereafter; training is finished, and data-mining algorithms may have low computational complexity depending on the volume of the data, which helps try to identify FDIA in a smart grid. Fuzzy logic-based techniques for spotting network breaches have also been developed. For example, the developers of [25] constructed artificial immune systems that recognize dangers like network flooding using fuzzy logic. Fuzzy logic is used to discriminate between illegal and legal traffic. The authors present a fuzzy logic-based technique for pinpointing jammer attacks. This serves as yet another example of how fuzzy logic can be used to identify cyberattacks. This method uses the precise channel evaluation, the low packet ratio, and the received signal intensity to ascertain if the connection loss was due to jamming. They had some perfect ideas for intermittent and persistent jamming.

Fuzzy logic was combined with other methods [26] to recognize different cyberattacks. Another crucial AI-based way is evolutionarily based algorithms. They are widely used for global advancement. Well-known evolutionary algorithms include genetic algorithms as examples. This kind of program can simulate how evolution and natural selection work. A genetic algorithm-based technique with two steps—training and detection—was proposed by the authors of [27]. They used a genetic algorithm in their research to remove all but the essential components of the detecting process. The authors conclude that this tactic works

well for various network intrusions. The authors of [28] examined the potential effects of genetic algorithms on various machine-learning approaches. The simulation results show that genetic algorithms and the other three machine learning methods can identify FDIA. Figure 3 shows different components of the smart grid.

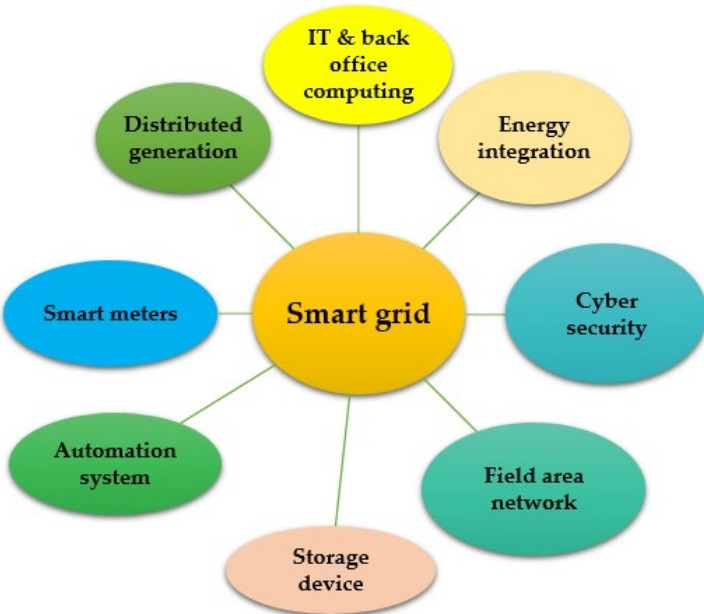

**Figure 3.** Essential components of the Smart grid [29].

Advanced metering infrastructure is essential to intelligent grid architecture. The primary purpose of AMI is to measure the energy consumption of integrated appliances and other devices, such as solar panels on roofs, gas meters, smart appliances, and water heaters. The smart meter, data concentrator, and central system are all constantly communicating with one another as part of AMI [30]. The meter data management system receives data from electricity meters via the AMI host system. MDMS is in charge of organizing and analyzing the data that utility systems send to it. Utilities and service providers can save costs and improve service quality due to the AMI system [31].

A Process Control System called SCADA enables the real-time monitoring, measuring, and analyzing data from the power grid. However, SCADA can also guarantee connections over short and long distances, making it ideal for installations [30]. The three main parts of this system are the Human Machine Interface, Master Terminal Unit, and Remote Terminal Unit [32]. There are three parts to the remote Terminal Unit. The first component has data processing capabilities, the second component has logic program execution capabilities downloaded from the MTU Master Terminal Unit, and the third component is primarily in charge of network configuration [33]. Another element of SCADA that assists in controlling and keeping track of the RTU is the MTU. The system's final element, the HMI, gives the SCADA operator a graphical user interface. Demand Side Management is a crucial part of the smart grid. This system regulates residential energy use. Demand Side Management can improve power market stability by balancing supply and demand [34]. Demand-side management has several benefits, including improved short-term reliability, lower peak-to-average demand and power supply ratios, cheaper user bills, and lower production costs. The stretcher of paper is shown in Figure 4.

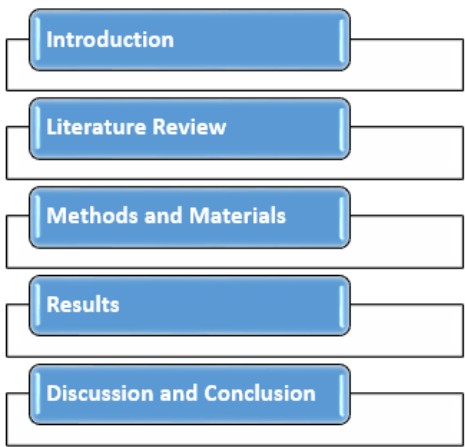

**Figure 4.** Structure of the paper.

Table 2 shows the list of abbreviations.

**Table 2.** List of abbreviations.

| Abbreviations | Full Form | Abbreviations | Full Form |
|---|---|---|---|
| P.M.U.s | power monitoring units | N.I.S | National Institute of Standards |
| A.I | Artificial Intelligence | A.M.I | advanced metering infrastructure |
| W.A.M.R | wireless asset management relay | IoE | Internet of Energy |
| S.G | Smart grid | E.I | Energy Internet |
| S.S | Smart system | IoT | Internet of Things |
| S.H | Smart House | D.O.E | Department of Energy |
| E.V | Electric Vehicle | EISA | Energy Independence and Security Act |
| I.G | Intelligent grid | NASPI | North American Synchro Phasor Initiative |
| N.E.T.L | National Energy Technology Laboratory | NERC | North American Electric Reliability Corporation |
| L.A.N | Local area network | EEGI | European Electric Grids Initiative |
| H.A.N | home-area network | ISGTF | Indian Smart grid Task Force |
| S.G.M.M | Smart grid Maturity Model | CPRI | Central Power Research Institute's |

## 2. Literature Review

In a multi-energy microgrid, numerous unknowns exist regarding the interactions between renewable energy sources, power demands, and electricity transaction costs. A two-stage, mixed-integer, deterministic, linear programming model of the problem has been developed, and it can be addressed by linearizing constraints and generating and reducing uncertain scenarios. The suggested approach is then tested on a microgrid that uses an IEEE 33 bus distribution network to control energy from various sources [34]. As smart grids replace conventional electrical grids, one of the significant problems that have developed is maintaining the system's safety. However, if the design and supporting infrastructure are created from the initial concept with security in mind, this problem can be solved. Therefore, implementing cyber security is a crucial and additional step. The National Institute of Standards and Technology initially recognized confidentiality, integrity, and availability as the three principles of smart grid security [35]. However, the authors highlighted the importance of accountability for smart grid security in Secrets that are frequently compromised when unauthorized people access private data.

On the other hand, integrity guarantees that data is sent without being changed or deleted. However, accessibility is a critical feature that ensures users access to the system's data in the context of smart grids. People cannot obtain information since it is not available [36]. Accountability assures that the system can be tracked and must be verified by a person, a device, or a government organization, which is essential for the security of the smart grid. Additionally, the recorded data can be used as proof in the event

of an attack to establish the actions taken by each user, including administrators, and to guarantee the accuracy of the data collected from each device [36]. Consequently, adopting the following four rules confidentiality, integrity, availability, and accountability, is the best way to safeguard smart grid systems. Smart grid networks are vulnerable to numerous attacks due to insufficient communication.

AI is widely used in the field of Cybersecurity. The digitization of manufacturing processes is usually correlated with machine learning, natural language processing, and robot-assisted process automation [37]. However, Cybersecurity has long used techniques of a similar nature. Consider the filtering system as an example of how machine learning might be helpful. It has been used since the early 2000s [38]. It is clear that methods have changed through time, and current algorithms can draw much more complex judgments. The digital security of smart grids has significantly improved due to recent AI developments. These improvements enhance the defenses against various threats. The five most common uses of machine learning are security (detection of fraud and viruses), privacy, business, and IT. Most people are unaware of how often artificial intelligence is used. Companies can quickly understand threats due to AI, which speeds up response times and ensures that best security practices are followed. Even while technologies like AI, 5G, and others are on the threshold of helping to resolve these problems, the energy sector must continue to invest to remain ahead of cyberattacks [39]. AI is also used to identify and stop intrusions into computer networks. Deep learning systems can also keep track of user identities if needed. Figure 5 describes the relationship between AI and Cybersecurity.

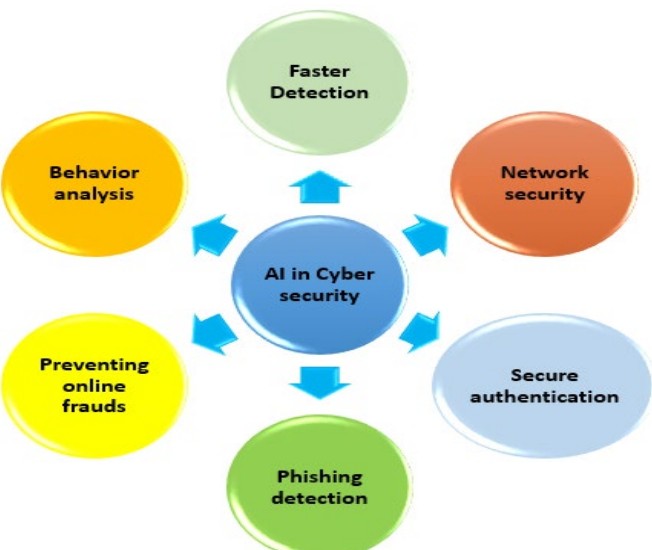

**Figure 5.** AI in Cybersecurity [40].

The use of databases infrequently or never, frequent location changes, access times, or other anomalies can all be picked up by AI algorithms [41]. Machine learning, in contrast, makes it easier to find data patterns that enable automated learning [42]. Utilizing cyber threat knowledge, smart grid users can quickly and effectively fix problems. Although today's security systems are perfect for identifying and stopping common threats, they cannot keep up with the growing need for Cybersecurity. None of these methods can contain zero-day vulnerabilities, an extremely slow cyberattack. A more flexible approach is needed to investigate data sets and find hidden security problems [43]. Machine learning has shown to be quite capable of identifying threats that were not there before using adaptive baseline behavior models. The security landscape would drastically change when predictive analytics and machine intelligence are combined with known and unknown data sets [44]. Table 3 illustrates how AI can be applied to strengthen security.

**Table 3.** Summary of AI Methods on smart grid.

| AI Technique | Advantages | Disadvantages |
|---|---|---|
| ANN | AI methods are more complex to understand than artificial neural networks. A multi-step process known as information technology is used to analyze data and look for a potentially unexpected pattern. It works with a range of teaching techniques [45]. | It has a higher computational cost and tends to overload. The model creation process is based on empirical research [45]. |
| SVM | Control parameters in ANN keep the model without being too accurate. This works best when there are apparent differences between the groups in the data set. The kernel technique makes it quick and easy to become an authority on a particular subject [46]. | Large data sets are too complicated for this method. Using this method when there are overlapping categories is not practical. Testing is a slow process [46]. |
| ANFIS | By combining the learning capabilities of an ANN with fuzzy systems, a neuro-fuzzy system may automatically create fuzzy if-then rules and optimize their parameters. This fixes the fundamental problems that have prevented designing fuzzy systems up to now [47]. | Depending on the number of fuzzy rules that were initially used. More calculations must be done as unclear regulations are added. |

One of the most popular ways to attack a smart grid is by jamming. An attacker can block communication by sending out constant or irregular signals. The operation of the smart grid network may be affected by various jammers [48], including continuous, random, misleading, and reactive jammers. Attacks known as "flow-jamming" use several jammers distributed throughout a network to slow down or stop normal traffic flow. Information is taken from the current network layer for these attacks. Jamming can be an extremely powerful strategy when used against a weak opponent. With centralized management, the jammer may be set to use just the right amount of power to stop a specific packet [49]. In a non-centralized jammer model, each jammer shares information with neighbor jammers to maximize efficiency. As a type of attack, spoofing attacks can be harmful to smart grid networks. These "spoofing" attacks fall under this category and include MAC spoofing, ARP spoofing, GPS spoofing, identity/data spoofing, and others.

A spoofed creates a fake grant in any of these attacks to deceive other nodes and damage the network's security, dependability, stability, and operation, which can compromise the integrity, confidentiality, and accountability of the smart grid [7]. Attacks can be launched against the network layer, the data link layer, and the physical layer. Injection attacks can happen when an attacker tries to remove, change, or add new data to a network, claim the authors of [50]. This might interfere with the smart grid's functionality and lead to a blackout. This cyberattack also corrupts data, compromises data integrity, and introduces malicious nodes into the network. Unlike earlier assaults, injection attacks might target the transport layer, the network layer, or the data-link layer [50]. A flooding attack is another hack that can be used against smart grid networks. This attack may limit system access at the network or application layer [51]. The target can expend all of its resources processing the fake messages sent to it. Another effect of this attack is that individual nodes cannot join the network. Man-in-the-Middle attacks on the smart grid are another type of cyberattack. The session and network layers are these intrusions' targets [52]. A man-in-the-middle attack happens in a smart grid when an attacker physically placed between two authorized devices connects to and sends communication between them. While the devices seem to speak, the attacker includes a third device in the conversation. These attacks' main goals are to interrupt network activity, change data while it is being transmitted, or obtain unauthorized access to sensitive data [53]. The security and privacy of a network may be risked if MITM is used. Social engineering is another cyberattack that could be used against smart grid technology. These attacks aim at the application layer and potentially risk the system's privacy [54]. According to the authors, social engineering is the greatest threat to information security. They explored social engineering techniques such as rob calls, phone/windows fraud, and reverse social engineering. Each of these attacks aims to trick victims into disclosing private information. These risks put users at

risk of having their personal information stolen for impersonation purposes, which can reduce their sense of security. A well-known passive attack on communication routes for smart grids is listening [54]. It goes after the network layer and affects the smart grid's specific privacy requirements. According to the attacks occur when a malicious user listens in on a conversation between two nodes on a LAN network to gather information. A user could use this sensitive data maliciously to interfere with the network. These assaults compromise the network's security.

A smart grid's physical and data link layers are known targets for timing-sensitive attacks [55]. The TSA is capable of managing, monitoring, and protecting large regions and 3-phase measuring devices. Synchronized measurements are required for numerous smart grid applications, and the vast majority of measuring instruments now come standard with GPS to provide accurate time information. These are vulnerable to spoofing attempts, just like other GPS-enabled devices. Smart grids require quick communication and control signals, making them more susceptible to cyberattacks such GPS spoofing and time-sensitive access [56]. By using hybrid brute force, reverse brute force, and credential stuffing, the presentation layer, session layer, or network layer can be compromised. Figure 6 shows the Cyber-Attack Classification.

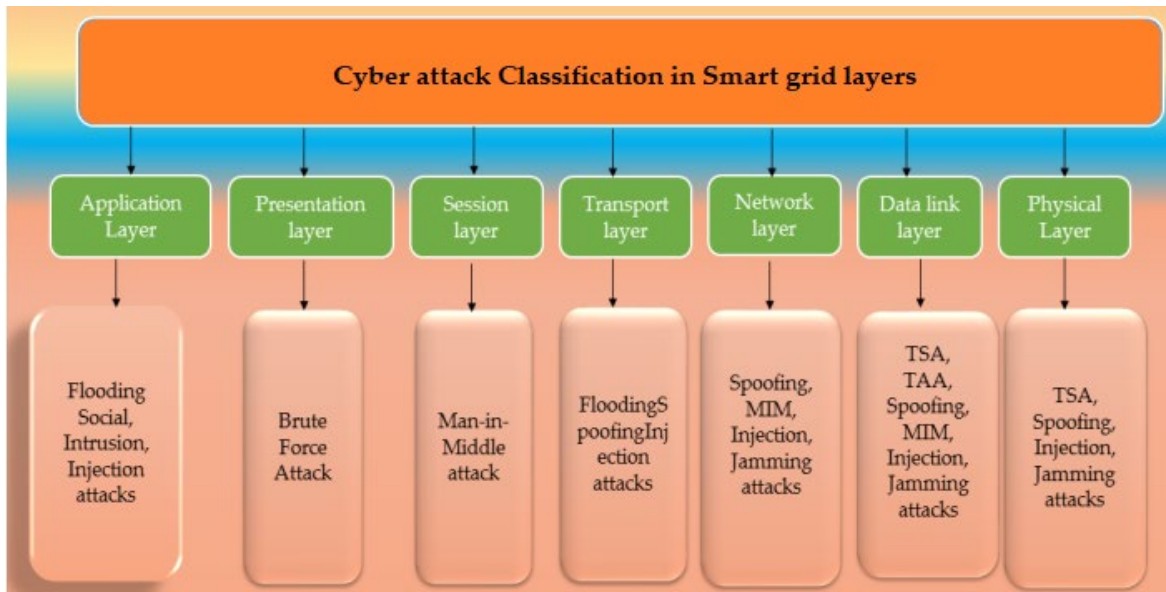

**Figure 6.** Cyber-Attack Classification Based on Communication Layers [10].

A "password guessing attack" is when an attacker attempts to guess or decode a user's username or passphrase to access the user's account or system. The authors of [57] explored the effects of attacks, including unauthorized access to the system and user accounts and the use of security flaws to reduce the system's privacy and dependability. An attacker can use a brute-force attack to get private data about smart grid users [24]. Another type of cyberattack on the smart grid is intrusions, in which an attacker takes advantage of flaws in the network to gain unauthorized access to nodes. Therefore, improper behavior, such as using force or making threats, may result in an invasion [58]. By interfering with the network's security and privacy at both the application and network layers, it also wants to waste network resources. The relevance and sensitivity of the smart grid make it especially vulnerable to intrusion attacks that could compromise the security of its network. Due to problems with authentication and integrity, modern SCADA systems, including smart grids, are becoming more vulnerable to cyberattacks like infiltration. Therefore, the network will function more effectively, and system downtime will be decreased if this attack can be located and halted. Traffic analysis attacks occur when an attacker listens to conversations and analyses what they hear. This attack aims to take over computers and other smart grid devices. The data connection layer is the target of this kind of attack [59].

Additionally, it may reveal confidential network data. In this attack, the assailant can listen in on conversations and analyze them to learn how network nodes converse with one another. Another well-known cyberattack on the data link layer of smart grids is the masquerade attack. This attack puts the security of the network's confidentiality, availability, integrity, and accountability at risk. To access a network or carry out illegal activity, an adversary could pretend to be an authorized user. To reduce the energy used by a home's electronic devices, an attacker usually alters a Programmable Communicating Thermostat in a smart grid [60]. Manipulating smart meters is one of the most common ways to undermine the smart grid. An attack at the physical layer can risk the security of a network. The information sent to any client can be changed in case of an assault on a smart meter. The consumer may pay more or less for electricity, depending on the results. Cyberattacks like buffer overflow, which require sending data to specific systems and components, are becoming more common in the smart grid. Concentrating on the application and transport layers also decreases network availability [61]. Because it could lead to a system crash and consume all network resources, this exploit should be avoided. Table 4 shows the Cyber-Attacks in Smart grids.

Another well-known smart grid vulnerability is the dummy attack. To attack the network layer, it makes use of network availability. The AMI network of the smart grid is penetrated by this attack, which takes advantage of a flaw in the Dynamic Source Routing protocol. As a result, storage space on our communication systems can become limited. One of the most noticeable effects of this attack is a 10–20% reduction in the number of packets that can be transmitted [62]. Targeting the smart grid in a hack known as an "IP spoofing attack" can also be used to decrease network accessibility. This kind of attack slows down and affects the person and the network's processing power in addition to hitting a single smart grid node. An attacker could use the broadcast address of the bounce site to deliver counterfeit packets from the source site. The bounce site may send incorrect packets to all hosts if it gets them. The approach can cause the target network to become overloaded. The network layer is the primary target of this kind of attack [63]. The HMI attack is a form of malicious online behavior that might result in a lack of the smart grid. In this case, the attacker uses a standard device attack (weaknesses in the operating system or software of the device) to get remote access to the server from their computer. The attackers' goal is to take total control of the machine that is being attacked. Infrastructure for smart grids and substations is managed and automated by SCADA devices, which could have security flaws. This attack necessitates little network expertise because the device's documentation is accessible. A hacker can easily take complete control of a compromised machine. The application layer's responsibility, availability, and integrity are all at risk [64].

Because it enables utility companies, customers, and producers to communicate automatically and in both directions via smart grid networks, advanced metering infrastructure has grown to be a critical part of the developing smart grid industry. Smart meters are high-tech devices that, in contrast to conventional meters, record a variety of information about a person's energy use, energy production, energy status, and diagnostics [65]. For purchasing, managing and watching user appliances, and troubleshooting, this data is really helpful. These data transfers all take place across a wide area network and are all kept in data centers that are hosted in the cloud. A centralized system may result in problems including a single point of failure, the potential for manipulation, and the loss of sensitive data. Performance, availability, and response time may be affected if more users connect to the same server. Smart meters and electric vehicles in smart grid systems also save a lot of information about payments and energy use [66]. These details and data are frequently disseminated to other businesses for monitoring, billing, and trading. Sharing a lot of data in such a complicated system; however, offers major privacy problems since middlemen, intermediaries, and trusted third parties might divulge private data on identities, locations, patterns of energy output and consumption, energy profiles, charging, or discharging quantities. The situation is made worse by the mistrust that exists between consumers and manufacturers. Because of this, it could be challenging for centralized parties to win

the trust of producers and customers by being truthful and open. It is a difficult effort to develop a decentralized AMI system that is dependable, private, and safe. Research on AMI and blockchain.

**Table 4.** Cyber-Attacks in Smart grids [14].

| Cyber-Attack | Objectives | Layers | Impacts | Security Requirements |
|---|---|---|---|---|
| Jamming Attacks | The main objective is to create trouble with both the data transfer and the data receiving. | Physical Data Link Networks | To prevent the sending and receiving of information collisions by blocking one or more nodes. | Availability |
| Spoofing Attacks | Trying to trick an authorized node into getting unauthorized access to the system | Physical Data Link Network Transport | Trying to mislead other nodes in the network. | Integrity Availability Confidentiality Accountability |
| Injection Attacks | The practice of inserting false or untrusted data packets into a network. | Data Link Network Transport Application | It injects false data perverting legal procedures and business activities with corruption the appearance in the network of nodes not authorized to be there. | Integrity |
| Flooding Attack | The main objective is to Bring about the loss and destruction of system resources. | Data Link Network Transport Application | In a network, failure of individual nodes and loss of availability of resources. | Availability |
| Man-in-the-Middle Attack | It blocks or alters the flow of data while it is being transmitted over the network. | Data Link Network Session | Access to confidential information that was not allowed. | Integrity Confidentiality |
| Social Engineering Attacks | Using fraud to encourage people to provide confidential information | Application | The users' right to privacy was violated. The system may suffer either temporary or permanent damage. Take confidential and sensitive information without permission. Theft of personal identity | Confidentiality |
| Eavesdropping Attack | Following up on and recording every bit of network activity | Physical Network | A violation of somebody's security | Confidentiality |
| Intrusion Attack | Acquire access to the node or network in an unauthorized manner. | Network Application | To Misuse the resources that are accessible on the network. | Integrity Confidentiality |
| Brute Force Attacks | Cracking user names and passwords requires a lot of work. | Session Presentation | It is obtaining access to a user's system or account without permission. | Integrity Confidentiality |
| Time synchronization Attack | Attacking the timing data and causing the nodes to lose their time synchronization | Physical | Events that compromise security, such as location estimation and fault detection, Performance decrease. | Integrity Availability |
| Traffic Analysis Attack | Execute command over the computers and other electronic devices linked to the network. | Data Link | Detect the message and analyze it to obtain information about the communication patterns between the nodes. | Confidentiality |

The authors in [67] offers a paradigm in which the authors use blockchain technology and smart contracts to improve the security and dependability of the smart grid. Both energy buyers and sellers will profit from the contracts' capacity to serve as a middleman. Productivity will rise, costs will drop, and the system will be safer as a result. After a transaction, a smart meter connected to the blockchain will submit the record, adding a new block to the distributed ledger with a timestamp that may be used to verify the data. The customer's bill can then be determined using the ledger information. The book's main issue is that it doesn't provide enough technical information.

In order to achieve decentralization and autonomy, a demand-side management paradigm for intelligent energy networks is described in [68]. This architecture creates a decentralized, secure, and autonomous energy network using blockchain technology, allowing each node to function independently of the others and the DSO. In addition, it is utilized to safely store the data blocks that smart meters collect about energy consumption. By establishing a prototype on the Ethereal blockchain platform using energy consumption and production traces from UK building databases, the method is finally assessed and confirmed. The findings show that this model is able to take into account different levels of energy flexibility and validate every demand response agreement in a manner that is almost real-time. Uncertainty exists over the energy profiles' anonymity in an open-source blockchain. The user can be identified by looking at transactions that are available to the public.

Security, privacy, and trust are three of any system's most important components. The similar level of security will be necessary for future intelligent grid systems [69]. This is sure that no unauthorized entity obtains information by putting in place the necessary cryptographic safeguards.

The most popular use of blockchain technology to date is Bitcoin. This is due to Nakamato's invention of a novel consensus technique in [70], which made it possible to create trust in distributed systems. A cryptographically secure data structure, a digital signature method, a time-stamp, and a numerous benefit are used in addition to the consensus process. Consensus mechanisms, for instance, are commonly used in blockchain applications to establish credibility. To handle fundamental security issues including privacy, integrity, authentication, authorization, non-repudiation, and anonymity, a variety of cryptographic approaches are used. It is not necessary to build a cryptocurrency in order to develop a blockchain-based decentralized system, even though coin applications are where the principles of consensus mechanisms and blockchain are initially exposed [9].

Nowadays, centralized platforms are used for a number of services by smart grid components such billing and monitoring, bidding, and energy trading. Although these technologies are advanced and work well, the existing smart grid system still has a number of important problems. As was already said, the smart grid also makes it possible to connect various RES, consumers, and cyber-physical systems. The grid's architecture is changing from a centralized, fully automated network to a decentralized, fully automated network as a result of the need for better interoperability. The EI idea is assisting in the transition in the smart grid industry from a producer-controlled network to a high-end decentralized network [68].

The decentralized nodes of the network all agree regarding what is happening, guaranteeing that the blockchain always works as intended.

Many times, the peers in this network are able to carry out tasks like approving new members and keeping the network running without the aid of a centralized authority. The blockchain's network capacity grows as more computers join it on its own. The blockchain is a decentralized network that is mostly controlled by its users, which explains this. The blockchain is a safe but unreliable network because nodes can connect with each other without the help of a reliable third party and because all data and transactions are encrypted asymmetrically [71].

Blockchain differs from earlier systems that demanded constant trust in those in authority.

The data in the blocks cannot be changed until a majority of users oppose it because blockchain technology uses cryptography and keeps a shared global record across all nodes.

The immutability of the blocks and the validity of the contents may both be independently confirmed by nodes on the blockchain network [72]. As a result, the blockchain's architecture is incredibly transparent and reliable. Any node on the network may check the legitimacy of the blocks with this level of openness without requiring access to confidential information.

## 3. Methods and Materials

### 3.1. Research Method

The literature on IoT Security studies has grown in recent years as more and more academics have developed an interest in the field. With the use of the AND OR search operators, we were able to find a vast amount of information that was relevant to topics like "IoT", Machine Learning", Deep Learning", threats, "cyberattacks", and "vulnerabilities". We have also included other terms like "blockchain", healthcare", and "Data Mining. ML and DL" in our search for a solution to the issue of IoT security breaches.

### 3.2. Exclusion and Inclusion

IoT and machine learning approaches were used as a keyword string to find publications in databases from the IEEE, Springer, Scopus, Google Scholar, A.C.M., Science Direct, and Wiley. These works include research on machine learning categorization, IoT security, and integration of health systems. Papers that were first chosen for review are peer-reviewed before being published. To better understand how machine learning works and how it might be used to improve IoT security, this research explores publications that concentrate on machine learning-based approaches. After the initial search, any papers found were discarded. We only looked at a few articles because the review aimed to set standards for machine learning research criteria and methodology. The committee did not even read the additional recommendations.

### 3.3. Objective of the Study

Our main objectives of the study are.

1.　To know about the smart grid and its security issues.
2.　To know about the different types of attacks on smart grid.
3.　To know about the different methods to overcome these issues.
4.　To know about the Open Issues, Challenges, and Future Research Directions.

### 3.4. Smart Grid Communication Challenges

The Smart grid Communication Challenges are explained below.

#### 3.4.1. Interference

For the smart grid to work, smart meters must be installed in homes and businesses. In the typical household, more and more technologies are becoming standard. Nowadays, H.A.N.s are almost ubiquitous in homes. Under conditions of dense distribution, Network Area Controllers and smart meters may interact. This might result in inaccurate readings from smart meters, endangering the system's stability. Power line harmonics may cause communication equipment on the smart grid to malfunction.

#### 3.4.2. Transmission of Data Rate

The smart grid's communication infrastructure is essential for various reasons, some of which are the collection and analysis of data and the distribution of instructions to the system's numerous nodes. On the other hand, the smart grid necessitates an abundance of real-time sensors as well as smart meters, both of which, when combined, generate a substantial quantity of data that has to be sent rapidly while maintaining its integrity. In

addition, the foundation for mutual comprehension has to be created. Because of this, the smart grid requires a network connection that is both reliable and secure.

### 3.4.3. Regulation

A wide variety of different parts come together to form the electrical Grid. The smart grid relies on the interplay of many other factors, each of which plays a specific role. A well-integrated communication channel network is crucial for adequately constructing such a system. This has resulted in a proliferation of global initiatives aimed at standardization and developing generally accepted standards. These efforts have the backing of various institutions, including the IEEE, the European Committee for Standardization, the American National Standards Institute, and the International Telecommunication Union.

## 4. Results

### 4.1. Cyber-Attacks and Security Risks

It is common to see attacks, including who's conducting them, which system vulnerabilities are being used, which security gaps are being targeted, and the outcomes of the attack s possible risks. These are all important considerations that need to be considered [73]. When there is a risk to the confidentiality, integrity, or accessibility of data, systems, or other resources, a security flaw occurs. Each cybersecurity event offers a different threat to an individual or organization's systems and networks. Commonly referred to as "malware," malicious software is computer code created to harm a user's computer, server, or network [74]. Malware can enter a system by taking advantage of a security hole, such as when a user accidentally installs spyware by opening a malicious attachment or visiting a compromised website. Usually, the system's actual user won't be aware that this malicious program is present. Malicious software can easily access a system since there are many different ways to do it. A user may be deceived into installing malware by accessing a fake version of a valid file, going to a website known to spread malware, or connecting to an infected system or device. Another situation is when someone views malicious websites and is deceived into installing them. Any computing device is vulnerable to being infected by malicious software. Cyberattacks can target process control systems like Supervisory Control and Data Collection systems, end users, servers, and the hardware that connects them. Like the people it hurts, malicious software comes in various shapes and sizes. Examples include bot executables, Trojan horses, spyware, viruses, ransomware, and worms. Unhealthy programmed are constantly evolving and getting more complex [75].

The most cost-effective way to make long-term savings is to install efficient controls at the system's boundaries. A detection and prevention system is one type of this technology (firewall, anti-virus software). Using a security barrier, administrators can limit user access to a protected internal resource. Despite these safeguards, it is still possible for someone to misuse their access credentials. The degree of the misbehavior will determine whether a corporation uses a punishment from its accountability policy. Regrettably likely, comprehensive security strategies, access control techniques, and accountability mechanisms won't work [76].

The idea behind the Internet of Things is that everyday things may communicate with one another and other computers via the internet without the need for human interaction. Fires, break-ins, overheating, and door locks that unlock as someone approaches can all be detected and prevented with the use of Internet of Things technology.

The Smart workspace system, which makes use of Telegram messengers and the on-hand AI Chabot, is made to make it easier for employees to use electronic devices at their workplaces. Remote management is possible for the office's technology [77]. Additionally, the Chabot can inform staff members whether a device can be turned on or off or remind them to turn on the fan if the temperature rises too high. By enabling workers to manage all office technology from a single internet-connected device, such as a smartphone or laptop, the Internet of Things and artificial intelligence in the workplace can help employees save money on utilities and time.

The number of people using the messaging programmed Telegram Messenger is growing. There are 62 million active Telegram users right now, 15 million DAU, and 1 million new users join every week [78]. Since Telegram Messenger can be used with or without a smartphone and can also be accessed through a web browser, many people use it every day to connect with family, friends, and coworkers.

The term "smart grid" describes a power system that makes use of sensing technologies, communication, digital control, information technology, and other field equipment to coordinate its current operations and improve the efficiency and responsiveness of the power grid. The Photovoltaic Generation System may be tracked and measured with the help of the Internet of Things, and the WSN in particular [79].

The Internet of Things has also been utilized in agriculture to find farming-friendly places so that the correct plants can be planted [80]. IoT is used in medical to track heart rate.

It is feasible to build Smart door locks with Mobile Backend as a Service and home automation and smart security systems with Low Cost Real-Time [81] using an ESP 8266, a straightforward and affordable Internet of Things key.

Researchers in a range of fields are interested in neuro-fuzzy systems because of their better learning and reasoning capabilities. In neuro-fuzzy systems, the representation of implicit information via fuzzy inference systems and the ability of artificial neural networks to learn from their experiences are merged. Because of the speed, accuracy, and difficulty of creating computers, researchers have thought about using soft computing techniques to characterize, forecast, and manage dynamic nonlinear systems. Fuzzy logic systems and artificial neural networks are examples of soft computing techniques. To address critical difficulties, a number of research and engineering sectors are starting to combine the two schools of thinking. An intelligent machine's ability to reason and draw conclusions can be greatly enhanced by fuzzy logic. Fuzzy logic describes qualitative yet flawed data, allowing machine learning to be symbolically expressed. Neural networks are used because they can learn, are reliable, and offer a lot of parallel to a system. The neuro-fuzzy system is a great place to start when trying to solve machine learning problems because it can represent knowledge and self-learn. The Takagi-Sugeno-Kang fuzzy inference method is the most effective way to represent nonlinear dynamic systems. As a "multimodal" technique, TSK system modelling can use linear sub models to show how a complex nonlinear dynamic system behaves as a whole. One of the most well-liked neuro-fuzzy methods is ANFIS. Regression, modelling, forecasting, and control have all used it. The ANFIS utilizes a fuzzy inference system of the TSK type on a 5-layer network design. The two types of parameters in ANFIS are assumption and consequence. The relationship between the two groups of variables is described using fuzzy if-then rules. The biggest problem with ANFIS is that it uses a lot of computer resources and frequently produces models that are unnecessarily complex for even the most straightforward problems. The accuracy and training time of standard neuro-fuzzy networks have recently increased due to recent developments in learning algorithms and network architecture. A neuro-fuzzy system needs the following qualities to perform well: Positive qualities include one that can learn quickly, adapt on the fly, continuously optimize itself to attain the minimum possible global error, and use the least amount of computing power possible. Because hybrid techniques are used to continuously good them, most neuro-fuzzy inference systems take a long time to learn. On occasion, it's necessary to manually change some parameters. On the other hand, overfitting and local minima are easily induced by diffusion learning methods. While the input weights and hidden layer biases are chosen at random and can be thought of as a linear system, the output weights of ELM are determined using a straightforward generalized inverse operation. as opposed to the norm. Most CPSGs rely on wireless communication, it is simple for enemies to target that channel. Information technology attacks are those that limit access to data. Classical Intricate attacks operate on communication networks such as cognitive radio networks and mobile Adhoc networks [82]. By blocking trusted routing, these attacks slow down the network by taking advantage of infected insider nodes. A faulty sensing node may post inaccurate channel sensing data following an attack, which

is advantageous to the node but harmful to more reliable nodes. Intricate attacks are typically used by enemies for two reasons. The main goal is to stop criminal damage, which happens when untruthful people claim a channel is empty when it's actually in use. The second goal is exploitation when sensing indicates that a channel is not being used. This happens when an attacker makes up a busy signal to try to use a channel exclusively. Attackers can increase the efficacy of their attacks by giving priority to these goals [83]. The flaw in the aurora generator was found by the Idaho National Laboratory. By using a series of improper control commands, the attacker tries to open and close the circuit breaker on a generator in this type of attack. The disconnecting of the generator from the utility grid is referred to as interruptions. When the system and generator lose synchronization the safety mechanism can react, the Aurora Assault's goal is to reclose the circuit breaker. The aurora attack alters the generator's electrical output and rotational speed, which causes physical damage. This is due to the safety features of the generator being purposefully delayed to prevent accidental tripping. Closing the circuit breakers could be harmful to the generator because of the difference in frequency and phase angle between the generator and the main grid. Which circuit breakers are most vulnerable to Aurora attacks can be determined using a score method using vulnerability rating variables. Modeling and research into the effects of an aurora attack on the PCC and synchronous generator breakers of the micro grid may be found in [84]. Sync-check relays, which were previously used to defend against aurora attacks, are not permitted according to the IEEE 1547 Standard because they have the potential to unintentionally turn a micro grid into an island. The authors showed that tripping a micro grid's main circuit breaker could cause harm to the synchronous generator. The retail industry has recently given demand-response technology, which can enhance the functioning of the electrical grid, more attention [85]. At its core, demand-response demands, response an incentive-based control system in which incentives are communicated through command signals. In [86] simulation of an attacker with the goal of increasing the gap between production and consumption by hacking the transmission channel and changing market prices using an assault time series made the attack considerably more potent. One-shot assaults, in which harmful code is inserted just once, are different from this kind of attack. In [87] Looked at attacks that might insert false pricing information at any point over an extended period of time. Attacks that occur frequently can lead to power imbalances that cause overproduction, financial losses, and poor power quality. The amount of damage caused by repeated strikes was calculated by the authors using a technique called "sensitivity analysis." The authors used a sensitivity function based on the z-transform to model the system's behavior when analyzing its behavior over time. Challenges with energy-exchange systems were looked at by [88]. The end-user network's controllers quickly receive a price signal from the active market, and the network rapidly transmits bid information back to the controllers. Hackers can access the data that is sent between a prosumer and a market agent. The pricing attack was made worse by the insertion of fictitious prices and quantities from prosumers due to the deployment of malware. These attacks caused the market clearing price to fluctuate, each prosumer used a different amount of energy, and the overall demand on distribution feeders decreased. The authors in [88] examined two types of attacks: one that aims to undermine the system's reliability by changing the bid price to extreme values and another that aims to make money over time by keeping the bid price within predetermined limits in order to avoid detection. Prosumers are aware of their maximum bid amount thanks to the service agreement. Signal manipulation can be used by an attacker to get around these restrictions, but their actions will be exposed. Frequency regulation is extensively used in connected power networks. Controlling automatic generation would serve as an example. It guarantees that power moves along the tie-line between control zones at the predetermined rates and that the system's frequency stays within safe bounds. AGC uses data from distant sensors to ascertain a region's frequency and power flow. This enables it to assess how well the area is regulated The ACE shows the discrepancy between the recommended configuration for power exchange and system frequency and the current

configuration. Every few seconds, the AGC generators use the ACE to determine the control instructions automatically. Only a few minutes' worth of measurement validation processes, like state estimation, are carried out, which is insufficient to support the second-level frequency required by AGC. Since there is no way to check or locate the accuracy of measurements, AGC is vulnerable to attack. AGC is highly automated and only needs occasional system administrators' maintenance. When damaged, it can quickly alter how the system works [89].

Some of the most frequent outcomes of malware entering a network include the following:

- It prevents essential network elements.
- To spy using malware itself, it installs extra harmful software.
- It receives information and has access to personal data.
- It interferes with some components, rendering the system unusable for users.

Malicious malware, known as ransomware, has users pay in return for keeping their files from being deleted or denied access. Trojan horses are the most dangerous kind of malware because they can seem to be helpful, popular software while attempting to access sensitive financial information. Such "drive-by" attacks are a common way for malware to spread. The user must act before these records are created. Users only need to visit a secure website for their PC to become silently infected [90]. A compromised user's computer transforms into an Iframe and sends the victim's browser to a malicious website under the attacker's control after being compromised. Phishing is using email corruptly or falsely, for example, by sending spam or phishing emails. The goal is to gain the victim's private information to be used maliciously to access their bank accounts. This extreme threat is frequently used as part of a more extensive operation to gain access to corporate or governmental networks. As a result, it is commonly used in conjunction with other strategies. A type of phishing called spear phishing targets particular people or organizations, including those working in government or military intelligence. Criminals can get private company information through these attacks, which they can use to steal money or carry out other crimes. Whale phishing is spear phishing that targets powerful people, like the CFO or CEO, to gain sensitive data. When an attacker can place himself between two participants in a transaction or conversation, they commit a man-in-the-middle attack or listen in on a discussion [91]. Man-in-the-middle attackers most typically employ the following entries:

- Public Wi-Fi that isn't secure when unauthorized users place their devices in between a visitor's device and the network.
- If an attacker's virus successfully infiltrates the victim's PC, they can install software to obtain the victim's secure information.

IoT is becoming more and more popular because it can be used for a wide range of tasks, including intelligent energy management and industrial automation. At various grid nodes, Internet of Things sensors are installed to guarantee that electricity is transferred efficiently and correctly. IoT-SG integration problems must be fixed for the network to operate as planned. A neuro-fuzzy smart grid energy monitoring system for the Internet of Things is used by the operator's backbone to gather and transmit the parameters of the prediction model. we assess the effectiveness of an SG power monitoring system that is based on the Neuro-Fuzzy Internet of Things. Both customers and energy providers can gain from better resource analysis and management. Artificial neural network and fuzzy systems are combined in the ANFIS to provide a model that incorporates the best features of each. It makes use of a method called "Takagi-Surgeon fuzzy inference". This structure's layers each carry out a certain task and produce an output after processing inputs. The hybrid model combines the iteratively approach and the least-squares method. Any inference system with outputs from linear or constant membership functions can be built using Surgeon-type systems. For modern grids to function properly, the electrical infrastructure needs to be intelligent. Because it addresses the problems that plagued earlier grids, SG is a better and more dependable grid. A power monitoring system that is

enhanced by the Neuro-Fuzzy Internet of Things. Systems for managing solar and wind energy are controlled by the ANFIS smart grid controller. Wind and solar power plants will be able to produce much more energy with ANFIS-based power management. Using load power, current, and voltage as inputs, a Neuro-Fuzzy notion for power monitoring based on the Internet of Things was constructed A network or service must be taken down to stop responding to valid requests via a denial-of-service attack. DDoS attacks typically target the servers of well-known organizations, including financial markets, news organizations, banks, and governments. SQL injection changes database data that shouldn't be accessible to users [92]. A website's search bar is regularly used for malicious ends. This is known as SQL injection. A security flaw that cannot be addressed or that programmers are unaware of is called a "zero-day exploit." Engineers must constantly be on the lookout for this vulnerability. DNS tunneling enables the transmission of non-DNS communications on port 53, including HTTP and other protocols. As a standard and authorized method, DNS tunneling is usually disregarded when used for illegal activities [93]. Attackers can transmit their traffic outside and cloak it as DNS to hide the data they transport via the Internet. Table 5 shows the Cyber-Attacks and Security Risks.

**Table 5.** Cyber-Attacks and Security Risks.

| References | Types of Attacks | Solution |
| --- | --- | --- |
| [94] | FDIA | A method based on data-driven ML to identify stealthy FDIA on state estimate. |
| [95] | FDIA | Consider the notion drift while analyzing historical data, and concentrate on the distribution shift. Dimensionality reduction and statistical testing of hypotheses are used. |
| [96] | SCA | The data are transformed into a lower-dimensional space using the KPCA approach. The KPCA-transformed data are inputted for the ERT's SCA assault detection system. |
| [97] | DoS | A multi-class classification technique used in the smart grid for anomaly detection. |
| [98] | Pulse, ramp, relay trip, and replay attack | Supervised machine learning and model-based mitigation for anomaly detection (AD). The robustness and detection accuracy of the ML model was boosted by physics and signal entropy-based feature extraction. |
| [99] | FDIA | A CPADS created using ML techniques, network packet characteristics, and PMU. Metrics. |
| [100] | FDIA | A new FLGB ensemble classifier and optimum feature extraction ensemble learning-based FDIA detection algorithm are used. |
| [101] | FDIA | Extreme learning machines create a classifier that can identify abnormalities brought on by FDIAs. |

Data streams from the control center may be altered by intruders, resulting in incorrect choices that put the whole system in danger. Despite using encrypted communication, the P.M.U. and PDC, two essential components of the smart grid, are still vulnerable to hackers.

Components like P.M.U.s may have problems if a dependable connection cannot be ensured, which makes these problems worse. The selected method of communication has to be able to overcome these obstacles to be successful. The other components must still be able to carry out their intended tasks, notwithstanding the safety precautions that have been put in place. Because of this, measurements made by P.M.U.s, for example, rely on time. These measurements ought to arrive at the data-gathering facility within two seconds. No time must be wasted when a new security measure is implemented. The interconnection of the many cutting-edge and complex technologies that make up the smart grid is another reason for worry—the synchronization of measurements with P.M.U. Data is made possible by the use of G.P.S. The efficiency and dependability of the measures may be compromised if the G.P.S. signal is hacked or interfered with. The measurements collected from the P.M.U. will be worthless due to incorrect time stamping. Figure 7 shows the different types of attacks on the smart grid.

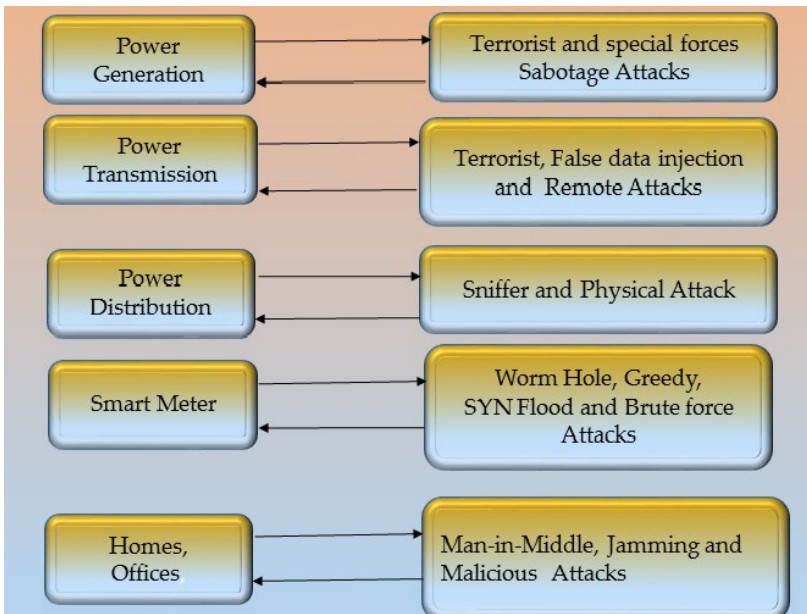

**Figure 7.** The possible attacks on the smart grid [102].

*4.2. ML and DL Algorithms for Cybersecurity*

As we'll see below, one of the most popular ways to overcome the limitations of traditional cybersecurity strategies is to use machine learning and deep learning algorithms. These methods can recognize intrusions that target the network in the issue. Machine learning is frequently seen as an essential part of cybersecurity because it can be used to attack and defend. One of these studies [103] looked at different machine-learning methods for identifying security flaws in IT systems. These techniques were random forests, support vector machines, naive Bayes, decision trees, artificial neural networks, and deep belief networks. The three main security challenges of intrusions, spam, and malware were the main areas of our examination.

4.2.1. Support Vector Machine Support Vector Machine

The usage of Support Vectors improves machine learning. The performance of numerous cybersecurity applications has been shown to benefit from the use of support vector machines. SVM is rarely used since it uses many resources, especially in real-time applications. Using kernel changes on the data, SVM establishes the ideal split between samples [104]. SVM transforms data using kernels to discover the best border between pieces. The authors in [105] created a model combining deep feature extraction with multi-layer support vector machines to identify abnormal behavior in a sizable amount of network traffic data. Distributed networks' security was ensured by doing this.

4.2.2. K-Nearest Neighbor

The K-Nearest Neighbor method uses a dataset's distance between two classes to assess their similarity or dissimilarity [106]. Since KNNs don't make assumptions, they can adapt to the numerous data formats now available more readily than other ML algorithms [107]. The decision tree is a supervised learning method in which the labeled dataset accurately predicts the model's output. The Wisdom Tree. A type of supervised learning known as decision trees uses labeled data to predict a model's production correctly. This machine learning method uses supervised learning and looks like a flowchart tree. To better prepare the large-scale cybersecurity dataset (UGR'16) for the anomaly detection model, used a decision tree and multilayer perceptron processing [108].

### 4.2.3. Deep Belief Network

According to one definition, a deep belief network comprises numerous layers, each of which can function as a restricted Boltzmann machine. Applications in the field of cybersecurity that require access to massive databases may find this helpful strategy. In [109], the writers thoroughly examined the use of deep belief networks and other deep learning techniques in cybersecurity. In the same way, the authors in [110] used the NSL-KDD dataset to assess the performance of the deep belief network for face recognition, pedestrian detection, and intrusion detection compared to a region extreme learning machine technique. In [111], Network performance may be monitored using traffic and payload parameters, enabling a secure deep neural network-based design. This framework was created to help identify hackers' behaviors in SCADA environments. A healthcare system's use of a blockchain-based architecture made it easier to pinpoint where unauthorized access attempts were attempted [112].

### 4.2.4. Recurrent Neural Networks

The directed graph structure of the recurrent neural network sets it aside from other neural networks. RNN also creates bidirectional signals and extends the network via loops. Since RNNs take longer to process than feed-forward neural networks, they are used less commonly in real-time applications. However, RNN was used to improve the accuracy of intrusion detection systems that used the dataset. In [113], To solve the issue of improper data injection in DC microgrids, a novel artificial intelligence-based method has been created. Researchers used RESs and a nonlinear auto-regressive external model to forecast dc voltages and currents. NARX aims to improve network performance compared to traditional RNNs in terms of speed, accuracy, and ease of understanding [114].

### 4.2.5. Convolutional Neural Networks

Compared to other deep learning algorithms, CNNs can learn from raw data. As a result, data extraction, which is generally done before training a model, is no longer necessary. Hidden networks, pooling networks, convolutional networks, and fully-connected networks are frequently included in convolutional networks (CNNs). In terms of cyber security, CNN lacks a particular leader in the field. The many security and privacy issues that organizations currently confront have led to the development of many CNN-based methods. For instance, as part of a cutting-edge method for identifying abnormal incursions, CNN was used to create a multiclass classification model for IoT networks. This process was used to find any possible threats [115]. The authors in [116] used this technique to find cyber-attacks on industrial control systems to create a small version of a wide range of industrial water treatment facilities. In [117], To recognize DoS attacks on IoT networks, writers used CNN. A distinct deep CNN technique was also recommended for malware identification [118]. It also allows the network to be used successfully on a GPU. A multi-CNN fusion technique was suggested to detect intrusion attacks on industrial IoT networks [119].

Thousands of sensors are being included in the smart grid's infrastructure to make the switch from a traditional grid to a smart grid. These sensors produce enormous amounts of data in the form of log files or time series data since they continuously check the health of the hardware to which they are connected. A smart grid system has several different kinds of sensors, including those that measure voltage, current, module temperatures, and irradiance. The information gathered by these sensors is processed before being sent to a server for storage. Both local and remote hosting options exist. The most secure way to store data is on a local server, but doing so limits the data's usefulness for identifying novel patterns or developing a deeper understanding of the subject of the study. The user has more control over how data is used when information is stored on a cloud server, accessible from a distance, and scraped to a computer using the GETS command. Machine learning approaches have lately been effective at locating cases of cyber intrusion. By examining past events, machine learning, on the other hand, may be able to identify intrusions. To

prepare for power outages, 54 linked J Ripper with Ada boost. The model divided the data into three categories based on its findings (assault, natural disturbances, and no event). An attack known as the false data injection attack is typical and has the potential to damage smart grid networks seriously. For utilities and consumers, tampering with data from smart meters might be pretty expensive. To locate the FDIA, researchers used ensemble-based machine learning [55]. Figure 8 shows the supervised learning process.

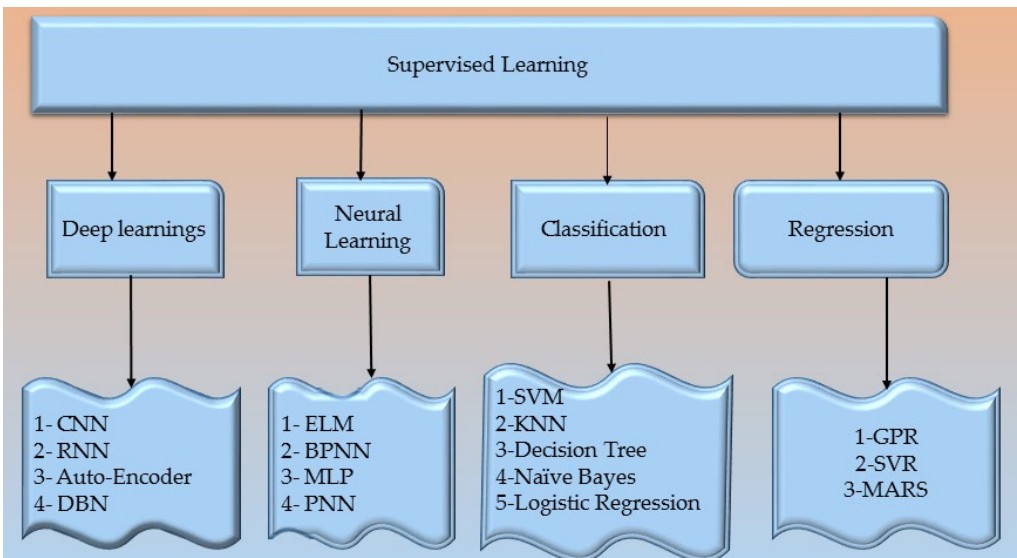

**Figure 8.** Supervised learning techniques in the smart grid.

The IEEE 14 bus system was used to evaluate the model. Unlike linear regression, naive Bayes, decision trees, and support vector machines, unsupervised ensemble models were more accurate than individual models, with the most incredible accuracy reaching 73%. The multilayer perceptron is used in [56] to examine how FDIA affects AI-based smart grids thoroughly. The study found that even if only 20% of the data is false, machine learning algorithms' accuracy might drop by 15%. This may significantly impact the decision-making process for the smart grid. Suppose a disruption happens, and the model cannot predict it because of inaccurate data, for example. In that case, data poisoning could cause the grid to become unstable and have unfavorable effects. The entire system can be negatively affected. The authors in [33] suggests using a conditional deep belief network technique to identify FDIA for power theft in real-time. The IEEE 118 bus and the IEEE 300 bus were used to test the model. The model's outcomes contrasted with support vector machines and artificial neural networks. Attacks that cause distributed denial-of-service to a smart grid are another possible danger. Attacks using distributed denial of service render servers and other crucial communication channels unusable. The goal of a DDoS assault is to bring down the targeted communication server by flooding it with fake requests. The authors in [57] proposes a multilevel auto-encoder model for detecting distributed denial of service attacks. An autoencoder has one input layer, at least one hidden layer, and one output layer. 49 characteristics and 700,000 data packets were used to train the model. These packets could be identified from others by their source IP address and ports, destination IP address and ports, both ends' jittering, record time, and type of attack. The UNSW-NB15 data set, available to the public for free, was used to develop the model. The results show that the auto-encoder-based prediction model performs better than the LSTM, random forest, naive Bayes approach, decision tree, k-nearest neighbor, and LSVM. Table 6 shows the Summary of different Machine Learning and deep learning Methods.

**Table 6.** Summary of different Machine Learning and deep learning Methods.

| References | Methods | Solution |
|---|---|---|
| [120] | Naive Bayes | Can be applied to analyses of both discrete and continuous variables. Features are assessed mutually exclusive, speeding up the process and making it applicable for real-time decision-making. |
| [121] | Support Vector Machines | In high-dimensional spaces, it effetely uses memory. Features that use numbers and categories |
| [122] | Decision Tree | Effectively uses memory in elevated environments. Features that employ categories and numbers |
| [123] | Sequential Pattern Mining | Frequent sequential patterns for a frequency support measure. |
| [124] | DBSCAN | Identify outliers and separate clusters of high density from sets of low density. |
| [125] | ADMIT | It doesn't need a lot of labeled data to function. Makes use of a recursive clustering algorithm, A K-means clustering variant. |
| [126] | A priori algorithm | As a result, the resulting restrictions make sense. Unsupervised, therefore labeled data aren't needed. |
| [73] | Radial Basis Function | Real-time network anomaly detection. |
| [127] | Random forest | Multi-class classification of network traffic threat |
| [128] | Extra-tree classifier | Multi-class classification of DoS, probe, R2L, and U2R |
| [129] | Radial Basis Function | Comparative classification between lazy, eager learning, and deep learning |
| [130] | Random forest | Comparative classification between lazy, eager learning, and deep learning. |
| [130] | Random forest | Android malware detection |
| [131] | ANN | Abilities to learn, classify, and process information; faster self-organization. |
| [132] | Deep Flow | Specifically designed to identify malicious software. Flow Droid, a program for static impurity analysis, is employed. Determines the paths taken by potentially sensitive data within Android applications |
| [133] | DBNs | Discovers layers of features and uses a feed-forward neural network to optimize discrimination. |
| [134] | Deep Belief Network | Real-time network anomaly detection. |
| [135] | Gated Recurrent Unit | Multi-class classification of network traffic threats |
| [136] | CNN-LSTM | Multi-class classification of DoS, probe, R2L, and U2R. |
| [137] | Deep Feed Forward | Differentiating between shallow, intermediate, and deep learning |
| [138] | Temporal convolutional networks | Comparative classification between lazy, eager learning, and deep learning. |
| [139] | CNN | Android malware detection. |
| [140] | Bi-LSTM | Classification of spam and ham from emails. |

Machine learning techniques such as unsupervised pattern discovery look for patterns in data without the help of labels. Although supervised learning algorithms have been the subject of decades of research, their use still depends on the users' access to the truth or their knowledge of the patterns to seek. This rarely happens when theory is applied in practice. Unsupervised learning can be used to find ways before data or predict what will happen in the future because it doesn't require labels. The method is, therefore, beneficial. Unsupervised neural networks can be used for several tasks, such as predicting load [63], determining stability [64], and detecting errors [65]. Auto encoders, variant auto encoders, and constrained Boltzmann machines are a few of these machines, but they are not the only ones. There are many examples of this. Clustering is a statistical approach by dividing a population or set of data points into subgroups that are comparable to the total. Untrained and uncontrolled persons carry out clustering. Some clustering methods include k-means, fuzzy c-means, hierarchical clustering, and DBSCAN. Additional clustering techniques not covered here exist. Applications are categorized using efficiency noise analysis. Data handling for smart grids typically use dimension reduction. Moving data from a high-dimensional space to a low-dimensional area is crucial to this method. These techniques have made it much simpler to utilize the information obtained [66]. Principal component analysis (PCA), linear discriminant analysis, extended linear discriminant analysis, and nonnegative matrix factorization is DR approaches used in smart grids [67]. The un-supervised learning process is shown in Figure 9.

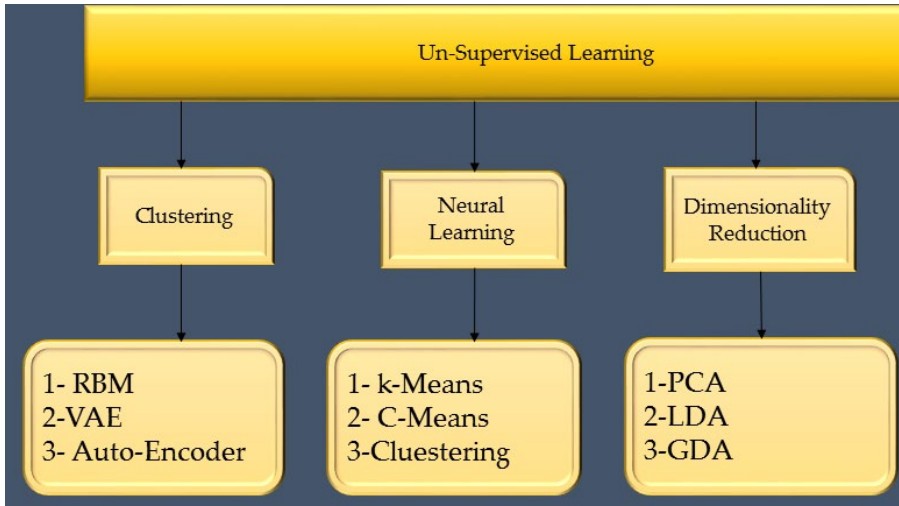

**Figure 9.** UN Supervised learning techniques in the smart grid.

### 4.2.6. Deep Reinforcement Learning

Reinforcement learning (RL) algorithms have the ability to maintain order in unpredictable situations. Therefore, the described POMDP challenge can be resolved using RL. The issue can be resolved using either a model-based RL algorithm for POMDPs [141] or a model-free RL algorithm without having to learn the underlying model. In general, only an unsubtle model can be learned using the model-based approach because it requires a two-step solution that is more challenging to compute. Attack-free worldwide anomaly identification methods include the Euclidean detector [142] and the arcos metric-based detector [26]. These systems compare expected and actual meter readings (using the Kalman filter) and, if the difference is greater than a set threshold, they declare an attack or anomaly. These detectors, however, only look at one sample at a time, so they cannot tell if attacks are taking place at the same time as strange results. Because of this, they are unable to differentiate between short-term anomalies generated, for instance, by an unfavorable system intervention and longer-term anomalies caused by system-level randomness. As a result, we need methods for universal attack detection that are more reliable than those that look for anomalies. Here, we look at the issue of smart grid security from the defender's point of view and use RL approaches to find an effective detection system [143]. The issue can also be viewed from the perspective of an attacker, in which case the goal is to determine the most harmful attack strategies. For vulnerability analysis, which is the process of identifying the worst possible thing an attacker might do to a system and then taking precautions against it, a challenge like this can be very helpful. RL has been the focus of numerous vulnerability investigations. For examples of FDI and sequential network topology attacks [50]. We also point out that the issue can be seen simultaneously from both the defenders and the attacker's points of view, just like in game theory. Multiage RL is a single-agent RL extension that heavily relies on game theory. This is thus because each actor's best behaviors depend on both their surroundings and the actions of other agents. Furthermore, stochastic games extend Markov decision processes to the multi-agent environment, where the game is played in a certain order, has many states, and is subject to payoffs that depend on the actions of all agents. To solve stochastic games, offer a number of RL-based techniques [144]. Additionally, if the environment's fundamental state, other agents' actions and rewards, etc. are only partially observable, the game is a partially observable stochastic one, which is often more challenging to solve.

### 4.2.7. Cloud-Based Detection and Mitigation

When combined with IoT technologies, cloud computing offers quick Internet access to a range of cloud services, including memory, storage, processing, network capacity, and database applications. One helpful feature of cloud computing is "pay as you go".

It is challenging for utility companies to build and execute this architecture to lower the cost of the hardware, software, and network services for the Smart grid. It is crucial to maximizing the network infrastructure's existing buffer, storage, constrained processing, and bandwidth since smart meters generate much additional traffic in the Smart grid. The authors of [60] examined how cloud computing features could be used to defend smart grids from DDoS attacks. The authors of [61] suggest using a cloud-based firewall to prevent DDoS attacks on smart grids. We created 250 Gbps of data for this experiment to simulate a distributed denial of service attack. According to the simulation results, the grid Open Flow firewall is not particularly slow. To ensure that only authorized users have access to cloud-based data, [62] proposes an attribute-based online/offline searchable encryption solution for smart grid applications. Figure 10 shows a cloud-Based Detection and Mitigation.

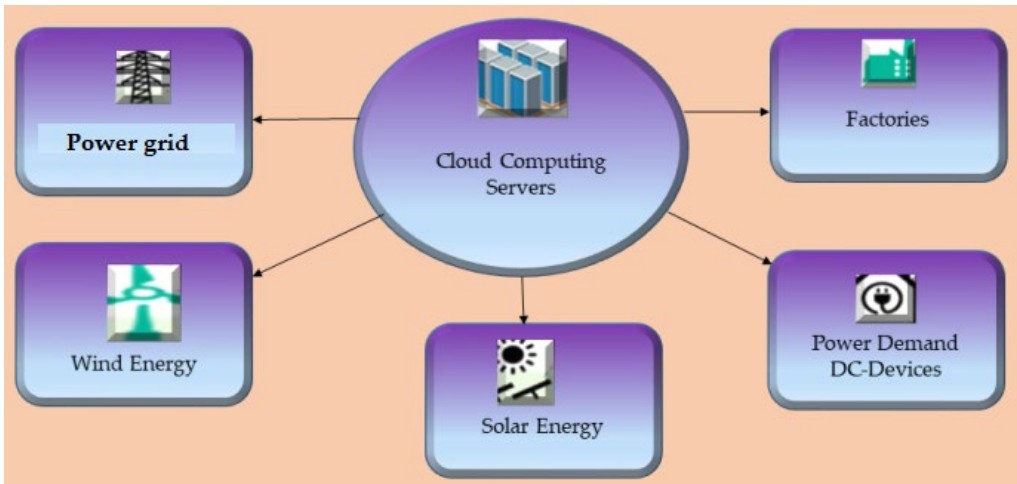

**Figure 10.** Cloud-Based Detection and Mitigation.

The authors of [63] describe a secure home area network that uses the cloud of things and is protected against threats, including brute force, replay, and capture. A model for assessing the security of a smart grid is created in [64]. A deep belief network comprises numerous RBMs, and a BP neural network is used to achieve this. To assess the overall level of security risks related to policy and organization and technological risks in general, SaaS, PaaS, and IaaS risks were looked at. Table 7 shows the Cloud-Based Detection and Mitigation.

**Table 7.** The Cloud-Based Detection and Mitigation.

| References | Objectives | Techniques | Limitations | Solutions |
|---|---|---|---|---|
| [145] | Auto-scaling of VM and VM-to-PM packing. | The approach is based on shadow routing. | Less no. of hosting PMs by intelligently packing VMs-into-PM. | Less no. of hosting PMs by intelligently packing VMs-into-PM. High performance in balancing the bandwidth utilization rate of hosts and sound management of both the physical and network resources. |
| [146] | Balance the load of network resources. | Layered virtual machine migration. | The migration cost is high. | |

**Table 7.** *Cont.*

| References | Objectives | Techniques | Limitations | Solutions |
|---|---|---|---|---|
| [147] | Minimize resource consumption and heavy traffic. | Cluster-aware VM collaborative migration scheme for media cloud. | The approach that has been proposed does not optimize the virtual machine migration in the media cloud. The expense of migration is costly. | A perfect migration is achieved by the utilization of clustering and placement algorithms, as well as an efficient migration of VM media servers. |
| [148] | Reduce energy consumption with high migration costs. | An improved grouping genetic algorithm (IGGA). | The migration cost is still high because of the migration of one VM at a time. | Increases the concentration score while bringing down the energy consumed while the consolidation score is high. |
| [149] | Minimize energy consumption and excellent migration cost. | Ant colony system (ACO) | The migration cost is still high because of migrating one VM at a time. | Reduces the overall amount of energy used by reducing the number of active PMs while ensuring compliance with the SLA's quality of service requirements. |
| [150] | Lessen energy consumption and excellent migration cost. | Firefly optimization approach. | Because migration may only result in a high utilization rate of network resources, the load cloud data centers are currently carrying is not going away. | Technique for migrating virtual machines in the cloud that is sensitive to energy consumption and moves overloaded VMs to regular PMs. |

### 4.3. Blockchain-Based Detection and Mitigation

The authors of [151] analyses each publication published between 2016 and 2022 that exclusively discusses protective measures for blockchain-based systems. The first cryptocurrency built on a blockchain, Bitcoin, was announced in 2008. The first blockchain-based cryptocurrency with smart contracts, Ethereum, made its debut in 2015. An alternate use of blockchain technology is the public blockchain project. blockchain technology was initially connected to the virtual currency bitcoin, but a new study suggests that it might be used for much more. Taking into account [152] claims, More investigation was done to determine whether blockchain technology might be used to improve cybersecurity. The authors looked into various potential fixes for blockchain security problems. To reduce the risk of cybercrime, a web-based cybersecurity awareness was developed. To maintain software security against hackers, the suggested method uses blockchain technology [112].

It was shown that a data-transfer system with an object-categorization algorithm might be created using blockchain technology due to its security; blockchain a sort of distributed ledger technology that has recently appeared as one of the most useful in numerous industries. On the blockchain, each block contains data, an index, a time stamp, a hash, and the hash of the block before it. A block cipher, in the opinion of many, forms the basis of blockchain dependability. Suppose the hash value of one block changes; all succeeding blocks in the chain must also change. Usually, it takes a lot of time and money to achieve this on a computer. According to the authors of [153], a policy architecture for data flow between autonomous system operations and agents that aren't performing their duties should be built using blockchain technology. These actions were all taken to combat the FDIA. Three sections make up the model: "data", "detection", and "blockchain". Figure 11 shows the blockchain Applications.

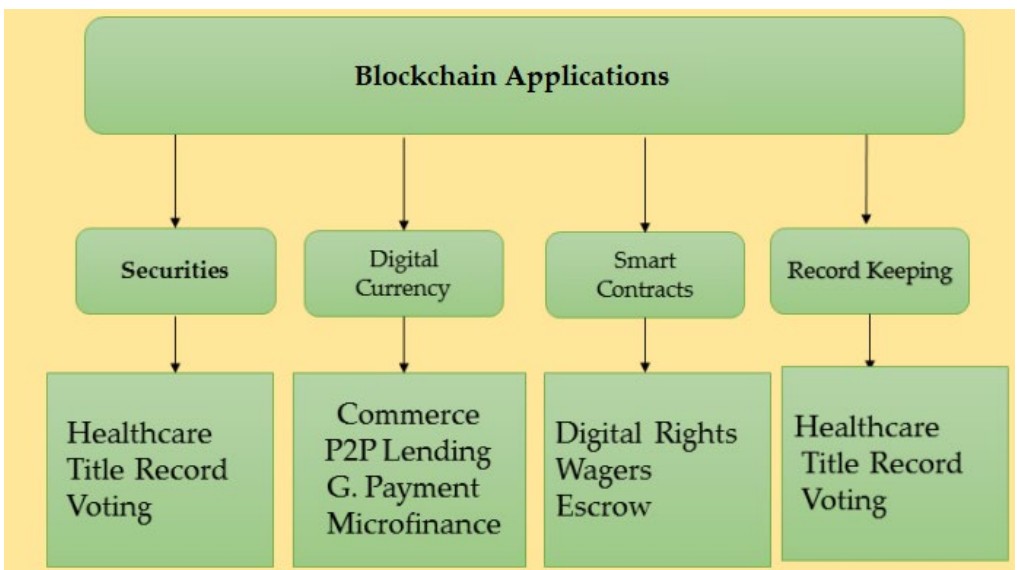

**Figure 11.** The blockchain Applications.

Information is gathered by the data layer and sent to the detection layer for community detection. The blockchain layer also secures the community detection and transaction record. In their article, the authors created a blockchain-based system for smart meters and service providers to communicate through encrypted communications [154]. The technique stops FDIA from happening on the smart meter's side. Smart meters serve as controller nodes in this study, starting all interactions with the service provider. Data is shared and validated throughout a network by auditing and broadcasting transactions. Service providers communicate with one another over P2P networking. A new transaction or block cannot be added until consensus has been validated, and only then can a new block be added. Each trade generates a unique key using the SHA-256 method. In this study, the authors showed that a framework built on a blockchain might be used to speed up data transmission and reception inside a P2P service provider network. This study [155] created a decentralized security paradigm using smart contracts and the lightning network in a blockchain environment. The registration, scheduling, verification, and payment processes are among the several procedures in this method. The authors of [156] created a product power system device design that combines hardware security with a blockchain-based method for maintaining a distributed security mechanism that checks the provenance of incoming communications. Table 8 shows the blockchain-Based Detection and Mitigation.

**Table 8.** Blockchain-Based Detection and Mitigation.

| References | Methods | Short Description | Findings |
|---|---|---|---|
| [157] | A game theoretic approach | A framework for energy trade and decision-making based on game theory | The strategy makes P2P trade both fair and optimum. |
| [158] | Networks of bilateral agreements for peer-to-peer energy trading | Networks for P2P energy trade that are bilateral and scalable | combines real-time and forward contract trading strategies |
| [61] | blockchain Applications in Smart grid | It looked at new blockchain applications and how they were used in the SG. | It showed the advantages of blockchain in the electrical network and the SG framework SPB, which reduces the costs, size, and processing time associated with energy trade. |

**Table 8.** *Cont.*

| References | Methods | Short Description | Findings |
|---|---|---|---|
| [159] | A Distributed Private Energy Trading Platform. | Presented a proof-of-concept for a secure private blockchain energy transaction system. | It showed the advantages of using blockchains in the electrical network and the SPB framework, which lowers the expenses, volume, and processing times related to energy trade. |
| [160] | Energy Trading Between Individuals Using a Virtual Power Plant Which Is Powered by Smart Contracts Stored on a blockchain | A public, sale price purchasing mechanism SC enables is recommended for energy trading. | Auction-based energy-trading platform |
| [161] | blockchain-based smart contract architecture for distributed generation trade and management | An infrastructure built on the blockchain to close the demand-response gap between energy supply by producers and consumer demand in peer-to-peer energy trading. | More than 25 individuals can trade energy at once due to it. |
| [162] | Blockchain-enabled Peer-to-Peer energy trading | investigates the best application of blockchain technology for peer-to-peer energy trading | The method is cheap for blockchain transactions. |
| [163] | Energy sharing between peers using batteries | It proposed an energy-sharing architecture based on energy pieces in a community market with a shareholder energy storage system, consumers, and users | Maximizes the income output for the energy supplier |
| [164] | Energy-backed token trading that is peer-to-peer and based on a decentralized blockchain platform for active producers and consumers | Utilize the blockchain to enable peer-to-peer trading of energy tokens | The suggested strategy ensures a global and practical resolution while requesting no private information from the participants. |

*4.4. Hardware-Based Security*

The smart grid system is useless without devices connecting to the internet. These devices must resist cyberattacks since they gather, process, and transmit data over the channel. The authors in [165] discussed some of the most critical hardware security issues. Security flaws might appear in many ways, including physical attacks, hardware Trojans, and side-channel analyses. The attacker wants to avoid being identified during the actual physical attack by the authentication procedure. System flaws were exploited using reverse engineering to plan the attack. An opponent can determine the cryptographic keys by analyzing the profile of numerous variables, including current, voltage, and frequency, using a method called "side-channel analysis". Any deliberate alterations or additions to a circuit are referred to as hardware Trojans. Hardware Trojans are malicious applications that steal sensitive information, change circuitry, or lower the system's dependability. According to the authors' analysis, path delay fingerprinting may be utilized to identify hardware Trojans. Smart meters, sensors, and communication devices are examples of IoT devices that battle with how much energy they can use and how little power they can use [166]. Because they allow fully secure authentication without requiring the device to have cryptographic expertise, PUFs are perfect for low-power Internet of Things (IoT) devices. Even so, by analyzing historical data and events, it is now possible to predict PUF behavior with a 95% degree of accuracy because of the development of machine learning [167]. To prevent machine learning-based attacks from breaching PUFs, the study's authors [167] developed a CTPUF, or configurable tristate PUF, using an XOR-based technique that hides the connection between the issue and the solution. The environment is too chaotic for the

machine learning model to detect recurring patterns between the challenge and responses. The results of this study showed that the accuracy of machine learning models that used CTPUF, such as support vector machines, artificial neural networks, and logistic regression models, was about 60%. Another study [168] used machine learning models to show the shortcomings of voltage-over-scaling (VOS)-based authentication. The studies also made abuse a possibility. The authors of this work developed a VOS technique that is immune to ML by fusing earlier challenges with keys. The results showed that when the challenge self-obfuscation structure was used, the ML model was approximately 51.2% accurate.

*4.5. Future Improvements and Challenges*

Source datasets are crucial for both cybersecurity and machine learning in a similar way. Most publicly available data is older, which might not be enough to identify trends in undetected cyberattacks. The most recent attacks and their repeated patterns remain a mystery, even though current data can be processed into knowledge through several different processing steps. As a result, it's possible that certain conclusions won't be drawn with exceptional precision using future processing or machine learning approaches. The production of numerous new cybersecurity datasets for particular problems like predicting attacks or detecting intrusions is a significant difficulty when using machine learning techniques in cybersecurity. Security datasets may be unbalanced, noisy, lacking crucial details, meaningless, or include examples of security vulnerabilities that are inconsistent with one another. The training of machine learning models may be more challenging and time-consuming when a dataset contains this kind of consistency [169]. These data concerns need to be fixed before using machine learning to create a data-driven cybersecurity solution.

Knowing everything there is to know about the issues with cybersecurity data is essential, as is finding reasonable solutions to these issues by using either current algorithms or brand-new algorithms to, among other things, locate malware and intrusions. One method to directly address these problems is feature engineering [170], which examines model features to remove related parts.

This method reduces the complexity of the data. A key strategy for dealing with measurement errors is using hybrid models, as explained in [73], or data creation, as described in [171]. More vulnerabilities that could lead to data leakage need to be addressed. The most popular and well-known techniques in cyber security use signatures to find intrusion attempts [172]. Due to data shortages, overly simplified characteristics, or inadequate profiling, these systems might overlook some assaults or events. These problems can be mitigated by signature-based or hybrid detection strategies combining signature-based and anomaly-based detection techniques. With a hybrid learning strategy that makes use of numerous machine learning techniques, intrusion detection, malware analysis, phishing detection, etc., all perform better. Machine learning, statistical analysis, and deep learning algorithms can be combined to make wise cybersecurity management decisions.

Due to the enormous amount of network traffic data and the high number of minute traffic features, a security model based on machine learning has frequently been questioned regarding its effectiveness and performance. Principal component analysis, singular value decomposition [73], and linear discriminant analysis have all been used by researchers to handle high-dimensional data [173].

Contextually, it might be advantageous to include low-level information in datasets that could be connected to problematic behaviors. This kind of contextual data may be categorized using an ecosystem or taxonomy for upcoming research. Therefore, choosing the best features or extracting the most important ones while considering machine-readable aspects and the context presents another challenge for machine-learning approaches in cybersecurity. To create effective cybersecurity solutions, this is necessary.

When models are used to produce predictions even while essential data is missing or significantly varies between datasets, this is known as data leakage [169]. Prediction models frequently result in too hopeful conclusions when they are being created. Still, when evaluated on new data, unsatisfactory findings list this problem, known as "leaks

from the future," as "one of the top 10 data mining defects." They suggest using exploratory data analysis to find and fix leakage sources. EDA enables machine learning models to collect more accurate data, enhancing the dataset's usefulness. Leakage detection and exploitation are substantial contributors, according to recent studies [174]. They were also noted as a critical factor in the failure of a data mining program. Researchers address the use of giveaway characteristics in data mining competitions to forecast the objective in another paper. This is because certain qualities were added afterward [175]. This article looks at the most popular techniques for categorizing documents and possible dataset structures for binary prediction. Each observation was given a "legitimacy tag" during the data collection phase, and data breaches were subsequently identified using a learn-predict separation. Maximum accuracy values of 91.2% for naive Bayes, 87.5% for k-NN, and 94.4% for centroid suggest that the suggested strategy is effective based on numerous categories.

The use of EDA to detect leaks is an exciting area for future research since it can be used in various situations when the machine learning scientist has little control over the data-gathering procedure. Homomorphic Encryption (HE) is regarded as a significant technological achievement by many cryptography experts [176]. HE gives unreliable third-party access to private data without disclosing anything. The encrypted data may end up on the user's computer or an unauthorized distant server, but not the decryption key. No information will be disclosed to unauthorized parties so the host can relax. HE can be used for various things, including cloud computing, financial transactions, and defense against quantum computing. HE can be used in a constrained or expansive way. During machine learning training, Fully Homomorphic Encryption (FHE) aids in maintaining the confidentiality of sensitive data. Shallow machine learning and deep learning substantially rely on domain data, which isn't always easy to come by for free [177].

Asymmetric encryption techniques were first thought to be straightforward for quantum computers to decrypt [178]. A pair of keys—one public and one private—are used for asymmetric key encryption. By multiplying two huge prime numbers, these keys are created. We employ large encryption keys to ensure the security of our data because it is challenging to factor in large prime numbers but simple to factor in small ones. A more labor-intensive approach for factoring such enormous prime integers is Shor's algorithm. The superposition quantum computing method may allow the factors to be discovered in a small portion of the time needed by a traditional binary computer. Elliptic curve algorithms and digital signature technologies like RSA and DES are weak points. Grover's method [73], based on quantum computing, claims that it will only take 185 searches to find the key to a 56-bit DES. Despite the existence of quantum computers, symmetric essential techniques like AES are still secure. Researchers are examining if these constraints may be overcome using mathematical and quantum methods. A quantum key distribution example is the BB84 protocol [179]. Lattice-based cryptography and other mathematical techniques are also being researched. Asymmetric encryption cannot be solved with quantum computing; however, using it as subroutine helps speed up machine learning [180].

This can drastically shorten prediction times for algorithms like SVM, where constructing a hyperplane and performing kernel modifications can take some time.

They may also be used for deep learning if they are well-designed. However, there are problems because quantum neural networks move in a straight path.

## 5. Conclusions

Transitioning from a traditional grid to a smart grid is complex and loaded with the inherent risks of testing out trying to cut technologies. Creating and maintaining an effective communication network architecture is one of the smart grid's most difficult tasks. In addition, creating a reliable physical architecture and keeping it up-to-date are challenging tasks. In this study, the communication infrastructure of the smart grid was studied, and future cyberattacks and defense strategies were taken into account. It would help if you never risked starting an attack because even a small one can have harmful effects. We think the people who use or operate the communication network are just as vulnerable

to attacks as the network itself. If the attacks are not successfully dealt with, they can turn into easy targets. It was suggested that security measures be put in place for smart grid clients, their communications network, and smart grid operators to build a reliable smart grid network. We took this action because we know that hackers target computer systems and their users and administrators. We considered many essentials before concluding this conclusion, including the nature of the attack, its scope, the individuals it affected, and the results it produced. Cyberattacks were also categorized according to the features of the attacks, such as the large areas that were compromised, the methods that were utilized to carry them out, and the measures necessary to establish dependable and efficient defenses. To successfully implement smart grid technology, network security must be addressed. However, previous studies have shown that their impact is minimal when assessing cyber-security solutions for smart grid networks. Therefore, this study completes the gaps left by earlier research by providing an in-depth description of potential smart grid attacks and assessing various security solutions.

In this research, we propose a layer-based classification of cyberattacks and a grading of these attacks regarding integrity, availability, confidentiality, and accountability. Finally, we highlight persistent issues that can guide future studies. Based on the results of this study, it is clear that there is a great need for novel approaches that may collectively resolve the complications associated with security issues in smart grid infrastructures without compromising the efficiency and usefulness of the network. For instance, "important regions impacted" refers to geographical locations essential to the network's functioning. Cyberattacks on smart grids are currently the focus of an extended investigation into the formation of a categorical classification. In this article, we will examine the many challenges that the sector is presently facing regarding cyber security, as well as the solutions that are currently accessible and the expected needs for future research. A comprehensive understanding of the types of security threats and assaults that smart grids are vulnerable to, as well as how these threats and attacks can be avoided, can be obtained by a review of the available research and literature. A comprehensive understanding of the types of security threats and assaults that smart grids are susceptible to and how these threats and attacks can be avoided can be attained by reviewing the available research and literature.

**Author Contributions:** Conceptualization, T.M. and I.H.; methodology, T.M. and H.M.I.; software, T.M. and H.M.I.; validation, T.M. and S.K.; formal analysis, T.M. and S.K.; investigation, T.M. and I.U.; resources, M.I. and T.M.; data curation, M.I. and T.M.; writing—original draft preparation, T.M. and I.H.; writing—review and editing, T.M. and I.H.; visualization, H.H., I.U. and I.H. All authors have read and agreed to the published version of the manuscript.

**Funding:** The authors thank the Natural Sciences and Engineering Research Council of Canada (NSERC) and New Brunswick Innovation Foundation (NBIF) for the financial support of the global Project. These granting agencies did not contribute to the study design and collection, analysis, and interpretation of data.

**Data Availability Statement:** Not applicable.

**Acknowledgments:** The authors thank Spectrum of Knowledge Production and Skills Development (Sfax) for giving access to its premises and for its logistical support.

**Conflicts of Interest:** The authors declare no conflict of interest.

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
