# Peer review of "Analysis of Cyber Security Attacks and Its Solutions for the Smart grid Using Machine Learning and Blockchain Methods"

_futureinternet, doi:10.3390/fi15020083_

Round 1

Reviewer 1 Report

This paper reviews the Impacts of Cyber Security attacks on Smart Grid, and this topic is relevant in the field and it addresses the gap in the cyber security. But the review can be more comprehensive and more details regarding the future work should be added. All the tables and figures should be improved to be clearer and straight to the point, the following should be conducted before its publication:

1:  Fig. 1 is not clear, please make it in high quality.

2: The title is a bit simple, please give more details.

3: Fig.2 is a bit simple and not beautiful, please replot it with more details.

4: The literature review is not enough for the following reasons:

  A: More references should be given to make your review more comprehensive.

     B: All the references should be better within the last 5 years, please try to make an update.

5: The 2.2-2.7 is a bit short and needs more explanations and reference support.

6: The related work is not comprehensive enough, please review the below research: In terms of the smart grid concept: https://doi.org/10.1016/j.apenergy.2019.02.085.

7: All the figures should be plotted better, currently, there are a bit unclear and not beautiful.

8: The conclusion should be revised with more details and future work should be added.

9: The whole language should be checked and revised.

Author Response

The responses to the reviewer's comments and suggestions are incorporated in the attached file.

Reviewer 2 Report

In this paper, the authors describe and illustrate the types of information security threats and attacks that smart grids may be subjected to. The authors need to clearly explain the use of paper review to discuss the information security issues of smart grids, as well as the use of actual cases for illustration.

Author Response

The responses to the Reviewer's comments and suggestions are incorporated in the attached file (reviewer report).

Reviewer 3 Report

This is an overview of cybersecurity issues using Smart Grids, I believe it should be resubmitted because of a rather poor style and structure. Of course the topic is relevant but the originality is very low: this is a review. The AI/Smart part isn't well discussed (at a keyword level). 

1.  First of all I found more than one similar articles, for example, this one published 10 years ago has the same title:

https://ieeexplore.ieee.org/abstract/document/6162722

The munuscript is not better than the published analogs, because this paper wrote more about historical data than on the actual Smart Grid problems.

2.  The Conclusions are very introductory.  The merits of their own research could be hardly found in Conclusions. This section should be cardinally improved.

3.  Some references are outdated but the authors have the right to use them.

Some references are rather far from the topic like this one: Determinants of Lecturer Performance to Enhance Accreditation in Higher Education...

Some references are incomplete.

The reference problems are minor compared to the other mentioned problems. Here I wrote the 4-star grade.

4. Please include any additional comments on the tables and figures.

The quality of fig. 1, 3-6 seems to be rather low. I am not convinced if they could be published in this form.

In table 4 the text between lines Jan-16 and Dec-15 should be better described. It is unknown what part of the text belongs to Dec-15. Other little problems could be mentioned.

5. The following part of this section contains too general or introductory notes:

>>> A thorough investigation into a thematic taxonomy of cyber-attacks on smart grids is now being conducted. This inquiry is still ongoing. Innovative research techniques were used throughout the whole investigation. In this article, we look at the many cyber security issues that the sector is now facing, the solutions that are presently out there, and the anticipated needs for future study.

The AI-based methods are only mentioned at a keyword level, and nothing more found. 

All intelligent components of this research should be better described. Otherwise the word SMART should be removed from the title. The set of the best analogs should be enlarged and better described.

Author Response

(The authors gave the same response as above.)

Round 2

Reviewer 2 Report

In this paper, the authors describe and illustrate the types of information security threats and attacks that smart grids may be subjected to.

Reviewer 3 Report

Main notes concerning the improvement of the proposed article. 1. What is the main question addressed by the research? This is an overview of cybersecurity issues using Smart Grids
2. Do you consider the topic original or relevant in the field? Does it
address a specific gap in the field?
Actually it is acceptable for publication but I believe it should be resubmitted because of a rather poor style and structure. Of course the topic is relevant but the originality is very low: this is a review. I wrote that the AI/Smart part isn't well discussed (few keywords are mentioned).
3. What does it add to the subject area compared with other published
material?
First of all I found more than one similar articles, for example, this one published 10 years ago has the same title: https://ieeexplore.ieee.org/abstract/document/6162722 The MDPI version is not better than the published analogs because I wrote in the review that they wrote more about historical data than on the actual Smart Grid problems. This could be improved.
4. What specific improvements should the authors consider regarding the
methodology? What further controls should be considered?
The authors wrote more about historical data than something concerning the actual Smart Grid problems. 
5. Are the conclusions consistent with the evidence and arguments presented
and do they address the main question posed?
Section Conclusions contains a lot of introductory notes. The merits of their own research could be hardly found in Conclusions. For pity, this part of my comments to the first version was neglected.
6. Are the references appropriate?
Some references are outdated but the authors have the right to use them. Some references are rather far from the topic like this one: Determinants of Lecturer Performance to Enhance Accreditation in Higher Education... Some references are incomplete. The reference problems are minor compared to the other mentioned problems. Here I wrote the 4-star grade.
7. Please include any additional comments on the tables and figures.
The quality of fig. 1, 3-6 seems to be rather low. I am not convinced if they could be published in this form. In table 4 the text between lines Jan-16 and Dec-15 should be better described. It is unknown what part of the text belongs to Dec-15.

Other little problems could be mentioned. I believe that I will have to write them when the new version is written. I am convinced that this version shouldn't be published.

Author Response

The revision file is attached

Round 3

Reviewer 3 Report

1. Please spell-check the article. 

For example, block chain with small B, typos in the modified figures, etc.

2. Different types of LSTMs have been mentioned but not included in Deep Learning classifications/figures. Meanwhile they usually outperform classical RNNs. 

3. Why Deep Reinforcement Learning systems are not mentioned? In Cybersecurity issues their role becomes more and more significant.

4. Why SMART security is described only at SCADA levels? Where are other Smart/Cyber Physical issues?

5. The blockchain part is a new and perspective addition to this multidisciplinary research. On the other hand, it stays rather isolated from the main part. Please improve its description. Does it affect the reactive features of the proposed interdisciplinary research (work in a real time, complexity issues, and so on)?

6. The mentioned fuzzy applications are interesting and perspective in all types of cybersecurity applications but the advice is to use different types of neuro-fuzzy systems (ANFIS, etc.)

7. It is a good idea to mention the role of intelligent chatbots in IoT security issues. 
